# TRUST REGIONS FOR EXPLANATIONS VIA BLACK-BOX PROBABILISTIC CERTIFICATION

## ABSTRACT

Given the black box nature of machine learning models, a plethora of explainability methods have been developed to decipher the factors behind individual decisions. In this paper, we introduce a novel problem of black box (probabilistic) explanation certification. We ask the question: Given a black box model with only query access, an explanation for an example and a quality metric (viz. fidelity, stability), can we find the largest hypercube (i.e., $\ell_\infty$ ball) centered at the example such that when the explanation is applied to all examples within the hypercube, (with high probability) a quality criterion is met (viz. fidelity greater than some value)? Being able to efficiently find such a *trust region* has multiple benefits: i) insight into model behavior in a *region*, with a *guarantee*; ii) ascertained *stability* of the explanation; iii) *explanation reuse*, which can save time, energy and money by not having to find explanations for every example; and iv) a possible *meta-metric* to compare explanation methods. Our contributions include formalizing this problem, proposing solutions, providing theoretical guarantees for these solutions that are computable, and experimentally showing their efficacy on synthetic and real data.

## 1 INTRODUCTION

Numerous feature based local explanation methods have been proposed (Ribeiro et al., 2016; Lundberg & Lee, 2017; Simonyan et al., 2013; Lapuschkin et al., 2016; Selvaraju et al., 2016; Dhurandhar et al., 2022; Ramamurthy et al., 2020; Dhurandhar et al., 2023) to explain individual decisions of black box models (Goodfellow et al., 2016). However, these methods in general do not come with guarantees of how stable and widely applicable the explanations are likely to be. One typically has to find explanations independently for each individual example of interest by invoking these methods as many times. This situation motivates the following question considered in our work:

*Given a black box model with only query access, an explanation for an example and a quality metric (viz. fidelity, stability), can we find the largest hypercube (i.e. $\ell_\infty$ ball) centered at the example such that when the explanation is applied to all examples within the hypercube, (with high probability) a quality criterion is met (viz. fidelity greater than some value)?*

Answering this question affirmatively has benefits such as: i) providing insight into the behavior of the model over a region with a quality guarantee, a.k.a. a *trust region* that could aid in recourse; ii) ascertaining stability of explanations, which has recently been shown to be important (Liao et al., 2022) for stakeholders performing model improvement, domain learning, adapting control and capability assessment; iii) explanation reuse, which can save on time, energy and even money (Dhurandhar et al., 2019); and iv) serving as a possible meta-metric to compare explanation methods.

Since we assume only query access to the black box model, the setting is model agnostic and hence quite general. Furthermore, note that the explanation methods being certified could be model agnostic or white-box. Our certification methods require only that the explanation method can compute explanations for different examples, with no assumptions regarding its internal mechanism. We discuss general applicability in Section 8. As such, our contributions are the following: 1) We formalize the problem of explanation certification. 2) We propose an approach called Explanation certify (Ecertify) with three strategies of increasing complexity. 3) We theoretically analyze the whole approach by providing finite sample exponentially decaying bounds that can be estimated in practice, along with asymptotic bounds and further analysis of special cases. 4) We empirically evaluate the quality of the proposed approach on synthetic and real data, demonstrating its utility.

## 2 PROBLEM FORMULATION

Before we formally define our problem note that vectors are in bold, matrices are in capital letters unless otherwise specified or obvious from the context, all operations between vectors and scalars are element-wise, $[[n]]$ denotes the set $\{1, ..., n\}$ for any positive integer $n$ and $\log(.)$ is base 2.

Let $\mathcal{X} \times \mathcal{Y}$ denote the input-output space where $\mathcal{X} \subseteq \mathbb{R}^d$. We are given a predictive model[1] $g : \mathbb{R}^d \to \mathbb{R}$, an example $\boldsymbol{x}_0 \in \mathbb{R}^d$ for which we have a local explanation function $e_{\boldsymbol{x}_0} : \mathbb{R}^d \to \mathbb{R}$ (viz. linear like in LIME or rule lists or decision trees) and a quality metric $h : \mathbb{R}^2 \to \mathbb{R}$ (higher the better, viz. fidelity, stability, etc.). Note that $e_{\boldsymbol{x}_0}(\boldsymbol{x})$ denotes the explanation computed for $\boldsymbol{x}_0$ applied to $\boldsymbol{x}$. For instance, if the explanation is linear, we multiply the feature importance vector of $\boldsymbol{x}_0$ with $\boldsymbol{x}$. We could also have non-linear explanations too, such as a (shallow) tree or a (small) rule list (Wang & Rudin, 2015). *Also for ease of exposition, let us henceforth just refer to the quality metric as fidelity (defined in eq. 14), although our approach should apply to any such metric.* Given the above and a threshold $\theta$, our goal is to find the largest $\ell_\infty$ ball $B_\infty(\boldsymbol{x}_0, w)$ centered at $\boldsymbol{x}_0$ with radius (or half-width) $w$ such that $\forall \boldsymbol{x} \in B_\infty(\boldsymbol{x}_0, w)$, $f_{\boldsymbol{x}_0}(\boldsymbol{x}) \triangleq h(e_{\boldsymbol{x}_0}(\boldsymbol{x}), g(\boldsymbol{x})) \geq \theta$. Formally,

$$\max \quad w \quad \text{s.t.} \quad f_{\boldsymbol{x}_0}(\boldsymbol{x}) \geq \theta \quad \forall \boldsymbol{x} \in B_\infty(\boldsymbol{x}_0, w). \tag{1}$$

We say that a half-width $w$ or region $B_\infty(\boldsymbol{x}_0, w)$ is *certified* if the constraint in equation 1 holds for $w$, and *violating* if not. Problem 1 is a challenging search problem even if we fix a $w$, since certifying the corresponding region is infeasible as the set is uncountably infinite. Moreover, we do not have an upper bound on $w$ a priori. Thus for arbitrary $g(.)$, given that we have just query access and a finite query budget, we can only aim to approximately certify a region. Our desire is that the proposed methods will correctly certify a region with high probability, converging to certainty as the budget tends to infinity, while also being computationally efficient. The latter is important as one might want to obtain such trust regions for explanations on entire datasets, which may be very large. Sometimes, we equivalently state we query $f(.)$ rather than querying $g(.)$ and computing $f(.)$.

## 3 RELATED WORK

Explainable AI (XAI) has gained prominence (Gunning, 2017) over the last decade with the proliferation of deep neural models (Goodfellow et al., 2016) which are typically opaque. Many explanation techniques (Ribeiro et al., 2016; Lundberg & Lee, 2017; Selvaraju et al., 2016; Sundararajan et al., 2017; Dhurandhar et al., 2022; Ramamurthy et al., 2020; Montavon et al., 2017; Bach et al., 2015) have been proposed to address this issue and appropriate trust in these models. However, it is unclear how widely applicable are the provided explanations and whether they are consistent over neighboring examples. In this work, we provide this complementary perspective where rather than proposing yet another explainability method, we propose a way to certify explanations from existing methods by finding a region around an explained example where the explanation might still be valid. This has benefits like those mentioned in the introduction, as well as possibly leading to more robust explanation methods as we discuss later. The need for stable explanations (Liao et al., 2022), possible recourse (Ustun et al., 2019) and even robust recourse (Pawelczyk et al., 2023; Maragno et al., 2023; Hamman et al., 2023; Black et al., 2021) further motivate our problem, where the latter methods try to find robust counterfactual explanations – not certify a given explanation – using white/black-box access. Our work also complements works in formal explanations (Ignatiev, 2020; Arenas et al., 2022), which try to find feature based explanations that satisfy criterion such as sufficient reason (or prime implicants) and are typically restricted to tree based models or quantized neural networks.

Another related area, adversarial robustness (Muhammad & Bae, 2022), also studies the problem of certification (e.g. Katz et al. (2017); Gehr et al. (2018); Weng et al. (2018a); Dvijotham et al. (2018); Raghunathan et al. (2018); Cohen et al. (2019); Tjeng et al. (2019); Chen et al. (2019)) but for the robustness of a single model to changes in the input, where no explanation is involved. Robustness certification can be seen as a special case of our problem where the explanation is a constant function. Within the robustness certification literature, randomized smoothing methods are more similar to our work in also using only query (a.k.a. black box) access to the model. However, they certify a smoothed version of the original model (Li et al., 2023), where an example is perturbed using Gaussian smoothing to facilitate robustness guarantees. We are not aware of robustness certification works that use only query access to certify the original model. Zeroth order (ZO) optimization

---

[1] One can assume one-hot encoding or frequency map approach (Dhurandhar et al., 2019) for discrete features.

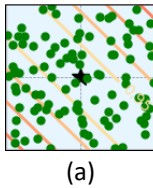 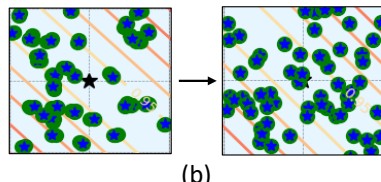 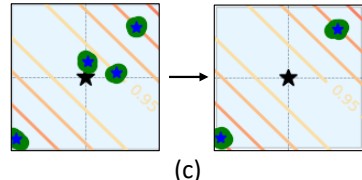

(a)  (b)  (c)

Figure 1: Illustration of our three certification strategies. (a) depicts one of the final steps of the unif strategy, while (b) and (c) depict two consecutive close-to-final steps of unifI and adaptI respectively. The setup is the same as in Section 7 with $d = 2$, $Q = 1000$. The boxes have width $w = 0.5$ which is the optimal width. The star in the center denotes the example whose explanation we want to certify, while the orange lines are level sets for fidelity ($\theta = 0.75$). The methods' different behaviors are apparent: unif queries examples uniformly at random, while unifI uniformly samples prototypes (blue stars) and then queries examples around these prototypes (green blobs). From one step to the next, unifI doubles the number of prototypes and halves the number of examples queried around each prototype. Contrastingly, adaptI, in the innermost loop, halves the number of prototypes where it adaptively queries more around prototypes close to low fidelity examples (lower left and upper right corners). Please see the uploaded videos to view the evolution of all steps for the three strategies.

methods have been proposed for adversarial *attacks* (Chen et al., 2017; Tu et al., 2019; Zhao et al., 2019), i.e., finding adversarial examples. In the experiments, we adapt a ZO method from a state-of-the-art toolbox (Liu et al., 2022) to our setting, and show that our proposed methods scale much better while still being accurate. In Weng et al. (2018b), extreme value theory is used for robustness certification, providing only asymptotic bounds for a special case of our problem (see Section 6.1). Moreover, their approach assumes access to gradients and is therefore not black box.

---

**Algorithm 1:** Explanation certify (Ecertify). (Code uploaded as supplementary file)

---

**Input:** example to be certified $x_0$, quality metric $f(.)$ (viz. fidelity), threshold $\theta$, number of regions to check $Z$, query budget per region $Q$, lower and upper bounding half-widths ($lb$, $ub$) of initial region, and certification strategy to use ($s = \{\text{unif, unifI, adaptI}\}$).

**Initialize:** $w = 0$, $B = \infty$

**if** $f(x_0) < \theta$ **then Output:** -1     # Even $x_0$ fails certification.

**for** $z = 1$ *to* $Z$ **do**
$\quad \sigma = \frac{ub-lb}{d}$     #Standard deviation of Gaussians in unifI and adaptI
$\quad (t, b) = \text{Certify}(lb, ub, Q, \theta, f(.), x_0, \sigma, s)$
$\quad$ # Find half-width of hypercube to certify.
$\quad$ **if** $t == True$ **then**
$\quad\quad w = ub$, $lb = ub$, $ub = \min\left(\frac{B+ub}{2}, 2ub\right)$
$\quad$ **else**
$\quad\quad B = \min\left\{|b_i - x_i| \text{ s.t. } |b_i - x_i| > lb \; \forall i \in [[d]]\right\}$, $ub = \frac{B+lb}{2}$

**Output:** $w$

---

## 4 METHOD

Our approach comprises Algorithms 1 and 2. Given the problem elements $x_0, f, \theta$ defined in Section 2, Algorithm 1 outputs the largest half-width $w$ that it claims is certified. Each iteration of Algorithm 1 decides which region to certify next based on whether the previous region was certified or not. The actual certification of a region happens in Algorithm 2, where we provide three strategies $s$ to do so. If Algorithm 2 certifies a region, defined by lower and upper bounding half-widths $lb$ and $ub$, then Algorithm 1 will either double $ub$ or choose it to be midway between the current $ub$ and $B$, where $B$ is an upper bound on half-widths to be considered. Otherwise, if the region is found to be violating, the next $ub$ is the midpoint between $B$ and $lb$, the width of the last certified region. Algorithm 1 will continue for a pre-specified number of iterations $Z$, after which it will output the largest certified region it found.

Some remarks on Algorithm 1: i) The lower bound $lb$ will typically be 0 initially, unless one already knows a region that is surely certified. If so, one could start from there. As we discuss later, if $g(.)$ is known to be Lipschitz for instance and the explanation function is linear, then one could also set a higher $lb$ value. ii) Although the end goal is to certify a hypercube around $x_0$, Algorithm 1 asks

---

**Algorithm 2:** $(t, \boldsymbol{b}) = \text{Certify}(lb, ub, Q, \theta, f(.), \boldsymbol{x_0}, s)$

---

Let $R = [\boldsymbol{x_0} + ub, \boldsymbol{x_0} - ub] \setminus [\boldsymbol{x_0} + lb, \boldsymbol{x_0} - lb]$ be the region to query in and let $q = \lfloor \frac{Q}{\log(Q)} \rfloor$

\# Choose sampling strategy as Uniform, Uniform Incremental or Adaptive Incremental

**if** $s == unif$ **then**

    Uniformly sample $Q$ examples $\boldsymbol{r_1}, ..., \boldsymbol{r_Q} \in R$ and query $f(.)$

    Let $\boldsymbol{r} = \underset{\{\boldsymbol{r_1}, ..., \boldsymbol{r_Q}\}}{\arg\min}\ f(\boldsymbol{r_i})$

    **if** $f(\boldsymbol{r}) \geq \theta$ **then Output**$(True, 0)$ **else Output:** $(False, \boldsymbol{r})$

**else if** $s == unifI$ **then**

    **for** $i = 1$ *to* $\lfloor \log(Q) \rfloor$ **do**

        Let $n = \min(2^i, q)$

        Uniformly sample $n$ examples (a.k.a. prototypes) $\boldsymbol{r_1}, ..., \boldsymbol{r_n}$ in $R$

        Sample $\lfloor \frac{q}{n} \rfloor$ examples (in $R$) from each Gaussian $\mathcal{N}(\boldsymbol{r_j}, \sigma^2 I)$ ($j \in [[n]]$) and query $f(.)$

        Let $\boldsymbol{r}$ be the minimum fidelity example amongst the $q$ queried examples

        **if** $f(\boldsymbol{r}) < \theta$ **then Output:** $(False, \boldsymbol{r})$

    **Output:** $(True, 0)$

**else if** $s == adaptI$ **then**

    **for** $i = 1$ *to* $\lfloor \log(Q) \rfloor$ **do**

        **if** $i2^i \leq q$ **then** $n = 2^i$, $k = i$ **else** $n = 2^k$

        Let $m = n$

        Uniformly sample $m$ examples (a.k.a. prototypes) $\boldsymbol{r_1}, ..., \boldsymbol{r_m}$ in $R$

        **for** $j = 1$ *to* $\lceil \log(n) \rceil$ **do**

            Sample $\lfloor \frac{q}{m\lceil \log(n) \rceil} \rfloor$ examples (in $R$) from each Gaussian $\mathcal{N}(\boldsymbol{r_k}, \sigma^2 I)$ where $\boldsymbol{r_k}$

            belongs to (selected) $m$ prototypical examples and query $f(.)$

            Find the minimum fidelity example (mfe) for each of the $m$ Gaussians

            **if** the mfe amongst these is $\boldsymbol{r}$ and $f(\boldsymbol{r}) < \theta$ **then Output:** $(False, \boldsymbol{r})$

            Otherwise, select the $\lceil \frac{m}{2} \rceil$ prototypes which are associated with the lowest minimum

            fidelity examples and set $m = \lceil \frac{m}{2} \rceil$

    **Output:** $(True, 0)$

---

Algorithm 2 to certify regions *between* hypercubes with half-widths $lb$ and $ub$. This is because the region with half-width $lb$ has already been certified at that juncture, and hence when certifying a larger region $ub$ we need not waste queries on examples that lie inside $lb$, instead saving the query budget for the region in between the two that has not yet been certified. We implement this by sampling examples from the larger hypercube and only querying those that lie outside the smaller hypercube.[2] iii) Other ways of updating the upper bound $B$ are discussed in Appendix F.

In Algorithm 2 we present three strategies: unif, unifI and adaptI. The first strategy, uniform (unif), is a simple uniform random sampling strategy that simply queries $g(.)$ in the region specified by Algorithm 1. If the fidelity is met for all examples queried then a boolean value of True is returned, else False is returned along with the example where the fidelity was the worst. In the second strategy, uniform incremental (unifI), we uniformly randomly sample at each iteration (i.e. from 1 to $\lfloor \log(Q) \rfloor$) a set of $n$ examples and then using them as centers of a Gaussian we sample $\lfloor \frac{q}{n} \rfloor$ examples. Again examples belonging to the region are queried and True or False (with the failing example) is returned. This method in a sense is performing a dynamic grid search over the region in an incremental fashion in an attempt to certify it. Our third, and possibly, most promising strategy is adaptive incremental (adaptI), where like in unifI we uniformly at random sample centers or prototypical examples, but then adaptively decide how many examples to sample around each prototype depending on how promising it was in finding the minimum quality example. So at each stage in the innermost loop we choose half of the most promising prototypes and sample more around them until we reach a single prototype or find a violating example. This method thus focuses the queries in regions where it is most likely to find a violating example. More intuition is provided in Figure 1 and Appendix D.

---

[2]we set $\sigma \propto \frac{1}{d}$ in Algorithm 1 since, with increasing dimension it becomes easier for an example sampled from a Gaussian to lie outside the hypercube as all dimensions need to lie within the specified ranges.

## 5 ANALYSIS

In this section, we provide probabilistic performance guarantees for Algorithms 1 and 2. We also verify that the total query budget used by each strategy is at most $Q$. Without loss of generality (w.l.o.g.) assume $\boldsymbol{x}_0$ is at the origin, i.e., $\boldsymbol{x}_0 = \boldsymbol{0}$. Then any hypercube of (half-) width $w$, where $w \geq 0$, can be denoted by $[-w, w]^d$, and $d$ is the dimensionality of the space. Let $f_w^*$ be the minimum fidelity value in $[-w, w]^d$, and let $\hat{f}_w^*$ be the estimated minimum fidelity in that region based on the methods mentioned in Algorithm 2. Note that we always have $f_w^* \leq \hat{f}_w^*$.

Given the above notation, the output of Algorithm 1 is a region $[-w, w]^d$ that is claimed to be certified, implying $\hat{f}_w^* \geq \theta$. However, the condition that we would ideally like to hold is $f_w^* \geq \theta$, involving the unknown $f_w^*$. Thus, we would like $\hat{f}_w^*$ to be close to $f_w^*$. In what follows, we provide bounds on the probability that $\hat{f}_w^*$ and $f_w^*$ differ by at most $\epsilon$, i.e., $P[\hat{f}_w^* - f_w^* \leq \epsilon] \geq 1 - p$, for any $\epsilon > 0$ and $p \in [0, 1]$.

One way to interpret our bounds is as follows:[3] Fix a value for $\epsilon$ and suppose that the region $[-w, w]^d$ is actually violating, by a "margin" of at least $\epsilon$: $f_w^* \leq \theta - \epsilon$. Then the probability that Algorithm 1 incorrectly certifies $[-w, w]^d$ ($\hat{f}_w^* \geq \theta$) is at most $p$. On the other hand, if $[-w, w]^d$ is truly certified, then $\hat{f}_w^* \geq f_w^* \geq \theta$ and Algorithm 1 also certifies the region. In the last case, if $\theta - \epsilon < f_w^* < \theta$, then $[-w, w]^d$ is violating but within the specified margin $\epsilon$ so we do not insist on a guarantee.

We note for our first result below that Algorithm 1 doubles or halves the range every time we certify or fail to certify a region respectively. Hence, to certify the final region $[-w, w]^d$ we will take $m = O(\log(w))$ steps. W.l.o.g. assume the number of subsets of $[-w, w]^d$ certified by the algorithm is $c \leq m$. Let $w_1 \leq \cdots \leq w_c$ denote the upper bounds ($ub$ in Algorithm 1) of the certified regions in increasing order, where $w_c = w$. Let us also denote the fidelity of an example $\boldsymbol{x}$ in between two hypercubes $[-j, j]^d$, $[-i, i]^d$ where $j \geq i \geq 0$ by $f_{j,i}^{(\boldsymbol{x})}$. The following lemma is a consequence of certification in a region being independent of certification in a disjoint region (all proofs in Appendix).

**Lemma 1.** *The probability that $\hat{f}_w^*$ and $f_w^*$ differ by at most $\epsilon$ decomposes over regions as follows:*

$$P[\hat{f}_w^* - f_w^* \leq \epsilon] = 1 - \prod_{i=1}^c P[\hat{f}_{w_i, w_{i-1}}^* - f_w^* > \epsilon] \geq \max_{i \in \{1, ..., c\}} P[\hat{f}_{w_i, w_{i-1}}^* - f_w^* \leq \epsilon], \quad (2)$$

*where $w_0 = 0$.*

From equation 2 it is clear that we need to lower bound $P[\hat{f}_{w_i, w_{i-1}}^* - f_w^* \leq \epsilon] \, \forall i \in \{1, ..., c\}$. Since the mathematical form of the bounds will be similar $\forall i$, let us for simplicity of notation denote the fidelities for the $i^{\text{th}}$ region by just the integer subscript $i$, i.e., denote $\hat{f}_{w_i, w_{i-1}}^*$ by $\hat{f}_i^*$ and similarly the fidelities for other examples in that region. We thus now need to lower bound $P[\hat{f}_i^* - f_w^* \leq \epsilon]$ for the three different certification strategies proposed in Algorithm 2.

**Uniform Strategy:** This is the simplest strategy where we sample and query $Q$ examples uniformly in the region we want to certify. Let $U$ denote the uniform distribution over the input space in the $i^{\text{th}}$ region and let $F_i^{(\text{u})}(.)$ denote the cumulative distribution function (cdf) of the fidelities induced by this uniform distribution, i.e., $F_i^{(\text{u})}(v) \triangleq P_{\boldsymbol{r} \sim U}[f_i^{(\boldsymbol{r})} \leq v]$ for some real $v$ and $\boldsymbol{r}$ in the $i^{\text{th}}$ region.

**Lemma 2.** *Given the above notation we can lower bound the probability of unif in a region $i$,*

$$P[\hat{f}_i^* - f_w^* \leq \epsilon] \geq 1 - \exp\left(-Q F_i^{(u)}(f_w^* + \epsilon)\right) \quad (3)$$

**Uniform Incremental Strategy:** In this strategy, we sample $n \leq q$ samples uniformly $\lfloor \log(Q) \rfloor$ times. Then using each of them as centers we sample $\lfloor \frac{q}{n} \rfloor$ examples and query them. Let the cdfs induced by each of the centers through Gaussian sampling be denoted by $F_i^{\mathcal{N}_{j,k}}(.)$, where $j$ denotes the iteration number that goes up to $\lfloor \log(Q) \rfloor$ and $k$ the $k^{\text{th}}$ sampled prototype/center.

**Lemma 3.** *Given the above notation we can lower bound the probability of unifI in a region $i$,*

$$P[\hat{f}_i^* - f_w^* \leq \epsilon] \geq 1 - \exp\left(-\max_{j \in \{1, ..., \lfloor \log(Q) \rfloor\}, k \in \{1, ..., n\}} \left\lfloor \frac{q}{n} \right\rfloor F_i^{\mathcal{N}_{j,k}}(f_w^* + \epsilon)\right) \quad (4)$$

---

[3] Another way is to regard $p$ as given and solve for $\epsilon$ to get a $(1-p)$-confidence interval $[\hat{f}_w^* - \epsilon, \hat{f}_w^*]$ for $f_w^*$.

The above expression conveys the insight that if we find a good prototype $r_{j,k}$ (i.e. close to $f_i^*$) then $F_i^{\mathcal{N}_{j,k}}(f_w^* + \epsilon)$ will be high, leading to a higher (i.e., better) lower bound than in the uniform case.

**Adaptive Incremental Strategy:** This strategy explores *adaptively* in more promising areas of the input space, unlike the other two strategies. As with unifI, let the cdfs induced by each of the centers through Gaussian sampling be denoted by $F_i^{\mathcal{N}_{j,k}}(.)$, where $j$ denotes the iteration number that goes up to $\lfloor \log(Q) \rfloor$ and $k$ the $k^{\text{th}}$ sampled prototype for a given $n$.

**Lemma 4.** *Given the above notation and assuming w.l.o.g. $F_i^{\mathcal{N}_{j,k}}(.) \leq F_i^{\mathcal{N}_{j,k+1}}(.)\ \forall j \in \{1,...,\lfloor \log(Q) \rfloor\}, k \in \{1,...,n-1\}$ i.e., the first prototype produces the worst estimates of the minimum fidelity, while the $n^{\text{th}}$ prototype produces the best, we can lower bound the probability of adaptI in a region $i$,*

$$P[\hat{f}_i^* - f_w^* \leq \epsilon] \geq 1 - \exp\left(-\max_{j \in \{1,...,\lfloor \log(Q) \rfloor\}} \left\lfloor \frac{(n-1)q}{n \log n} \right\rfloor F_i^{\mathcal{N}_{j,n}}(f_w^* + \epsilon)\right) \tag{5}$$

We see above that we sample exponentially more around the most promising prototypes (see Lemma 4 proof in the Appendix), unlike the uniform strategies which do not adapt. Hence, in practice we are likely to estimate $f_w^*$ more accurately with adaptive incremental especially in high dimensions.

**Remark:** It is easy to see (when the cdfs $F_i(f_w^* + \epsilon) \neq 0$) that for all the three methods asymptotically (i.e., as $Q \to \infty$) the lower bound on $P[\hat{f}_i^* - f_w^* \leq \epsilon]$ approaches 1 at exponential rate for arbitrarily small $\epsilon$, which is reassuring as it implies that we should certify correctly a region given enough number of queries. Moreover, $F_i(f_w^* + \epsilon) = 0$ is unlikely to happen in practice as can be surmised from proposition 2 in the Appendix. Now we can also lower bound equation 2 for each strategy.

**Theorem 1.** *Based on Lemmas 1, 2, 3 and 4 we have,*

$$P[\hat{f}_w^* - f_w^* \leq \epsilon] \tag{6}$$
$$\geq \begin{cases} 1 - \min_{i \in \{1,...,c\}} \exp\left(-Q F_{w_i}^{(u)}(f_w^* + \epsilon)\right) & \textit{unif} \\ 1 - \min_{i \in \{1,...,c\}} \exp\left(-\max_{j \in \{1,...,\lfloor \log(Q) \rfloor\}, k \in \{1,...,n\}} \left\lfloor \frac{q}{n} \right\rfloor F_{w_i}^{\mathcal{N}_{j,k}}(f_w^* + \epsilon)\right) & \textit{unifI} \\ 1 - \min_{i \in \{1,...,c\}} \exp\left(-\max_{j \in \{1,...,\lfloor \log(Q) \rfloor\}} \left\lfloor \frac{(n-1)q}{n \log n} \right\rfloor F_{w_i}^{\mathcal{N}_{j,n}}(f_w^* + \epsilon)\right) & \textit{adaptI} \end{cases}$$

We also have the following proposition regarding the number of queries used by each strategy.

**Proposition 1.** *unif, unifI and adaptI query the black box at most $Q$ times in any call to Algorithm 2.*

## 6 BOUND ESTIMATION AND SPECIAL CASES

In Section 5, we derived (with minimal assumptions) finite sample bounds on the probability of estimated and true minimum fidelities being close, which is directly related to correct certification. However, the cdfs $F_i(.)$ are generally unknown. In Section 6.1, we discuss the estimation of our bounds, and we also provide alternative asymptotic bounds that are cdf-free. In Section 6.2, we first provide a partial characterization of $F_i(.)$ for a piecewise linear black box and then discuss settings where the trust region can be identified even more efficiently using our strategies.

### 6.1 BOUND ESTIMATION

**Cdf $F_i(.)$ Estimation:** An attractive property of our bounds is that the cdfs are all one-dimensional, irrespective of the dimensionality of the input space. Hence, it is efficient to estimate the corresponding (univariate) densities. More specifically, given fidelity values sampled in a region by any of the three strategies, one can estimate a distribution of these fidelities using standard techniques such as kernel density estimation. The only challenge is that the point of evaluation $f_w^* + \epsilon$ is unknown because $f_w^*$ is unknown. We propose using $\hat{f}_w^*$ or $\theta$ as proxies for $f_w^*$. Since the bounds depend only on subsets that are certified, if we assume that these certifications are correct (i.e. $f_w^* \geq \theta$), then using $\hat{f}_w^*$ should provide (somewhat) optimistic bounds (since $\hat{f}_w^* \geq f_w^*$) while, $\theta$ will provide conservative ones. As we shall see in the experiments, both proxies are close.

Table 1: Synthetic results for $x = [0]^d$, $Z = 10$, $\theta = 0.75$, explanation is slope 0.75 hyperplane and optimal half-width is $\frac{1}{d}$. Standard errors for the half-width ($w$), bounds computed using Theorem 1 and EVT bounds for unif and unifI are in Tables 2, 5 and 6 respectively in the Appendix.

| $d$ | $Q$ | unif | | unifI | | adaptI | | ZO$^+$ | |
|---|---|---|---|---|---|---|---|---|---|
| | | $w$ | Time (s) | $w$ | Time (s) | $w$ | Time (s) | $w$ | Time (s) |
| 1 | 10 | 1 | .001 | 1 | .001 | 1 | .001 | 1 | .012 |
| | $10^2$ | 1 | .006 | 1 | .004 | 1 | .002 | 1 | 1.221 |
| | $10^3$ | 1 | .055 | 1 | .041 | 1 | .026 | 1 | 1.724 |
| | $10^4$ | 1 | .53 | 1 | .418 | 1 | .189 | 1 | 1.641 |
| 10 | 10 | .06 | .001 | .037 | .001 | .142 | .001 | .3 | .012 |
| | $10^2$ | .082 | .003 | .06 | .007 | .08 | .003 | .1 | .125 |
| | $10^3$ | .09 | .036 | .085 | .049 | .11 | .044 | .1 | 1.354 |
| | $10^4$ | .1 | .363 | .117 | .615 | .1 | .551 | .1 | 14.944 |
| $10^2$ | 10 | .012 | .001 | .006 | .001 | .007 | .001 | .05 | .031 |
| | $10^2$ | .012 | .005 | .007 | .012 | .008 | .005 | .025 | .3 |
| | $10^3$ | .011 | .054 | .009 | .158 | .01 | .09 | .012 | 4.072 |
| | $10^4$ | .01 | .632 | .01 | 1.692 | .01 | .51 | .009 | 55.87 |
| $10^3$ | 10 | $5 \times 10^{-4}$ | .003 | $3 \times 10^{-4}$ | .004 | $5 \times 10^{-4}$ | .002 | .037 | .307 |
| | $10^2$ | $6 \times 10^{-4}$ | .011 | .001 | .073 | $6 \times 10^{-4}$ | .044 | .012 | 2.579 |
| | $10^3$ | $8 \times 10^{-4}$ | .077 | .001 | 1.074 | $8 \times 10^{-4}$ | .511 | .003 | 28.335 |
| | $10^4$ | .001 | .588 | .001 | 13.786 | $9 \times 10^{-4}$ | 5.097 | .001 | 288.523 |
| $10^4$ | 10 | $6.3 \times 10^{-5}$ | .012 | $5.1 \times 10^{-5}$ | .098 | $5.8 \times 10^{-5}$ | .021 | .006 | 3.76 |
| | $10^2$ | $6.6 \times 10^{-5}$ | .072 | $7.7 \times 10^{-5}$ | 1.187 | $7.8 \times 10^{-5}$ | .43 | .004 | 34.602 |
| | $10^3$ | $8.3 \times 10^{-5}$ | .771 | $8.4 \times 10^{-5}$ | 12.452 | $8.5 \times 10^{-5}$ | 7.91 | $8.4 \times 10^{-4}$ | 391.494 |
| | $10^4$ | $8.9 \times 10^{-5}$ | 4.83 | $9.1 \times 10^{-5}$ | 112.58 | $9.4 \times 10^{-5}$ | 88.342 | $9.3 \times 10^{-5}$ | 4384.76 |

**Asymptotic (Cdf-free) Bounds:** Rather than finite sample bounds that depend on cdfs $F_i$, one could instead take an asymptotic ($Q \to \infty$) perspective and obtain results that are free of $F_i$. Extreme Value Theory (EVT) (Smith, 2003) is useful in this regard. Given our setting where the minimum fidelity $f_i^*$ in each region $i$ is finite, we can assume $F_i(f_i^* + \epsilon) \approx \eta \epsilon^\kappa$ as $\epsilon \to 0$ for some $\eta > 0$, $\kappa > 0$ as is standard in EVT (Smith, 2003). This would apply to all three strategies. We state an explicit asymptotic result for the case of i.i.d. fidelity samples, as it naturally follows from EVT. This i.i.d. case covers the unif strategy and an i.i.d. version of unifI discussed in the Appendix. Here in addition to the empirical minimum fidelity $\hat{f}_i^*$, we also use the second-smallest empirical value, denoted as $\hat{\hat{f}}_i^*$. Then the result of de Haan (1981) (also re-derived in De Carvalho (2011)) implies the following.

**Corollary 1.** *For the unif and i.i.d. unifI strategies, as $Q \to \infty$, we have*

$$P\left[\hat{f}_i^* - f_i^* \leq \epsilon\right] = \left(1 + \frac{\hat{\hat{f}}_i^* - \hat{f}_i^*}{\epsilon}\right)^{-\kappa}. \tag{7}$$

Corollary 1 is reminiscent of Lemmas 2–4 except that the region-specific minimum $f_i^*$ has taken the place of the overall minimum $f_w^*$. However, the two coincide for $i = i^* \in \arg\min_i f_i^*$. For $i = i^*$, the right-hand side of equation 7 is a valid lower bound on the probability $P[\hat{f}_w^* - f_w^* \leq \epsilon]$ in Theorem 1, as we discuss further in the Appendix. In our experiments, we estimate $i^*$ using the empirical minimum fidelities as $\hat{i} = \arg\min_i \hat{f}_i^*$. The exponent $\kappa$, as argued in de Haan (1981), can be taken to be $\kappa = d/2$, and thus the bound is completely determined given the fidelity samples.

## 6.2 SPECIAL CASES

**Characterizing cdfs $F_i(.)$ for a piecewise linear black box:** In Appendix B we provide a (partial) characterization of the cdfs $F_i(.)$ for piecewise linear black box functions, which cover widely used models such as neural networks with ReLU activations (Hanin & Rolnick, 2019), trees and tree ensembles, including oblique trees (Murthy et al., 1994) and model trees (Gama, 2004). This characterization assumes a linear explanation function and a commonly defined fidelity function (Dhurandhar et al., 2022; Ramamurthy et al., 2020).

**More Efficient Certification:** In Appendix C we discuss how having a black box model that is Lipschitz or piecewise linear can further speed up our methods. In the Lipschitz case we can automatically (i.e. without querying) certify a region and set a non-trivial $lb$ value with additional speedups possible. In the piecewise linear case instead of a head start (i.e. higher $lb$) we could stop our search early.

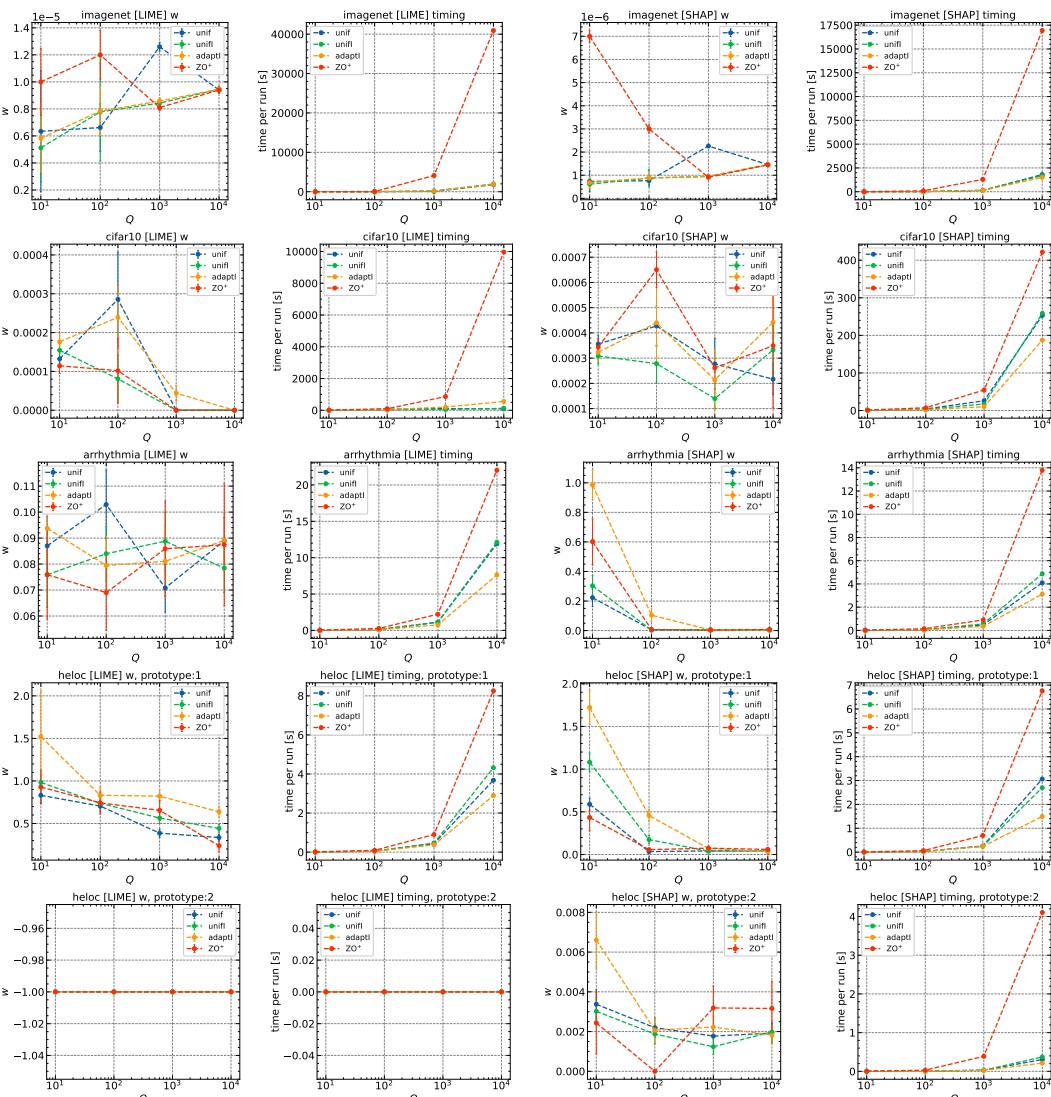

Figure 2: Each row corresponds to a dataset (Row #: 1-ImageNet, 2-CIFAR10, 3-Arrhythmia, 4,5-HELOC). First two columns are LIME half-width and timing results, while the last two columns are the same for SHAP. Our methods are significantly faster than ZO$^+$, while still converging to similar $w$ in most cases. It seems unif, unifI and adaptI are best for low (100s or lower), intermediate ($\approx 1000$) and high dimensionality (10000s) respectively. Trusting the converged upon half-widths, one can also compare XAI methods as discussed below.

# 7 EXPERIMENTS

We perform synthetic and real data experiments to verify the accuracy and judge the efficiency of our methods. For real data we additionally report interesting insights that can be obtained by finding trust regions. Since the problem setup is novel there aren't existing baselines in the explainability literature. Nonetheless, we adapt a ZO toolbox (Liu et al., 2022) to be usable in our setup. We refer to this method as ZO$^+$, where our Ecertify algorithm calls this toolbox as a routine similar to our three strategies. In all the experiments the quality metric is fidelity as defined in eqn. 14 (in Appendix), results are averaged over 10 runs, $Q$ is varied from 10 to 10000, $Z$ is set to 10, $\theta = 0.75$ (in the main paper) and we used 4-core machines with 64 GB RAM and 1 NVIDIA A100 GPU.

**Synthetic Setup:** We create a piecewise linear function with three pieces where the center piece lies between $[-2, 2]$ in each dimension, has an angle of 45 degrees with each axis, and passes through the origin. The other two pieces start at $-2$ and $2$ respectively and are orthogonal to the center piece. The example we want to explain is at the origin. We vary dimensions $d$ from 1 to 10000. In the main

paper we report results for the explanation being a hyperplane with slope $0.75$ passing through the origin. The optimal half-width is thus $\frac{1}{d}$. Other variations are reported in the Appendix.

**Real Setup:** We experiment on two image datasets, namely ImageNet (Deng et al., 2009) ($224 \times 224$ dimensions) and CIFAR10 (Krizhevsky, 2009) ($32 \times 32$ dimensions), and two tabular datasets, HELOC (FICO, 2018) ($23$ dimensional) and Arrhythmia (Vanschoren et al., 2013) ($195$ dimensional). The model for tabular datasets is Gradient Boosted trees (with default settings) in scikit-learn (Pedregosa et al., 2011). For ImageNet we used a ResNet50 and for CIFAR10 we used a VGG11 model. We also consider arguably the two most popular local explainers: LIME and SHAP. To have a more representative selection of examples to find explanations and half-widths, we chose five prototypical examples (Gurumoorthy et al., 2019) from each dataset. We show one to two examples for each dataset in the main paper where the others are in the Appendix. More details such as explainer settings, certification strategy settings (etc.) for each dataset are in the Appendix.

**Observations:** From the synthetic experiments, we see in Table 1 that although all methods converge close to the true certified half-width, our methods are an order of magnitude or more efficient than $ZO^+$. Also they seem to converge faster (in terms of queries) in high dimensions ($100$ to $10000$ dimensions). Comparing between our methods it seems unif is best (and sufficient) for lowish dimensions (up to $100$), while unifI is preferable in the intermediate range ($1000$) and adaptI is best when the dimensionality is high ($10000$). Thus the incremental and finally adaptive abilities seem to have an advantage as the search space increases. Although we query (at most) $Q$ times in each strategy, adaptI and unifI are typically slower than unif because we sample from $n$ different Gaussians $\log(Q)$ times as opposed to sampling $Q$ examples with a single function call. This however, will not always happen if a violating example is found faster, as seen for adaptI on some real datasets in Figure 2. We also report our Theorem 1 bounds (estimated as in Section 6.1) on the probability of closeness and the (additional) time to compute them in Table 5 in the Appendix. As can be seen the bounds converge fast to $1$ especially for adaptI and are efficient to compute (at most a few minutes). EVT bounds based on Corollary 1, shown in Table 6, are also high enough to be meaningful for unifI and improve with increasing $Q$, but become looser with increasing input dimensionality.

From the experiments on real data, we again see from Figure 2 that our methods are significantly faster than $ZO^+$, while they still converge to (roughly) the same half-widths in most cases. The running times are especially higher in the LIME image cases because LIME has to create masks for each image we certify. We also observe that adaptI is generally faster in most cases because it finds the violating examples faster that the other strategies. As such, in terms of convergence of the estimated half-width with increasing number of queries balanced against efficiency, we observe that unif is probably best for HELOC and Arrythmia which are low dimensional datasets, unifI is best for CIFAR10 which has dimension close to $1000$, and adaptI is best for ImageNet which has $40K+$ dimensions. Probability bounds are reported in Table 7 for ImageNet. Here again like in the synthetic case we see fast convergence especially for adaptI with the bound computation being efficient.

Interestingly, our analysis can also be used to compare XAI methods. We observe that LIME widths are typically much larger than those found for SHAP, and hence the explanations are more generalizable beyond the specific example. This however, does not mean that LIME is always more robust than SHAP as the quality of the explanation depends on the desired fidelity. SHAP typically has fidelity of $1$ at $x_0$, while LIME may have lower fidelity at $x_0$ but generalizes farther in the sense of fidelity remaining above the threshold. For instance, in row 4 in Figure 2 LIME has a fidelity of $0.87$ which is greater than our set threshold of $0.75$. The explanation here considers AvgMlnFile and NumSatisfactoryTrades as important factors, while SHAP considers ExternalRiskEstimate as the most important factor. The latter is more informative for the specific example but doesn't generalize as well nearby. The last row shows the downside of generalizability where LIME fidelity even for the example we want to explain is lower than our threshold of $0.75$ and so we return $-1$, but SHAP produces a trust region. Thus, one could select which method to use based on the desired threshold for the quality metric. As such, this type of analysis can be used to compare and contrast XAI methods on individual examples, on regions, as well as on entire datasets, and across different models.

## 8 DISCUSSION

Please see Appendix G for discussion on extensions and interesting future directions.

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

## A  PROOFS FOR RESULTS IN SECTION 5

*Lemma 1 proof.* To prove the first equality, we recall that $\hat{f}_w^*$ is the minimum of the fidelities sampled from $[-w, w]^d$, while $\hat{f}_{w_i, w_{i-1}}^*$ is the minimum over the samples restricted to the $i$th region $[-w_i, w_i] \setminus [-w_{i-1}, w_{i-1}]$. Since $[-w, w]^d$ is the disjoint union of these regions, it follows that

$$\hat{f}_w^* = \min_{i \in \{1, \dots, c\}} \hat{f}_{w_i, w_{i-1}}^* \tag{8}$$

and

$$P\big[\hat{f}_w^* > f_w^* + \epsilon\big] = P\left[\hat{f}_{w_1, w_0}^* > f_w^* + \epsilon, \dots, \hat{f}_{w_c, w_{c-1}}^* > f_w^* + \epsilon\right].$$

Since samples in different regions are independent, the joint probability above factors to yield the equality in the lemma (after taking the complement).

The inequality in the lemma follows from bounding all but the smallest of the $P\big[\hat{f}_{w_i, w_{i-1}}^* > f_w^* + \epsilon\big]$ factors by 1. The inequality is tight if the $\arg\min$ corresponding to equation 8 is a single region,

$$i^* = \arg\min_{i \in \{1, \dots, c\}} \hat{f}_{w_i, w_{i-1}}^*,$$

and $\epsilon$ is small enough so that $f_w^* + \epsilon < f_{w_i, w_{i-1}}^*$ for $i \neq i^*$ (recall that $f_{w_i, w_{i-1}}^*$ is the true minimum fidelity in the region $[-w_i, w_i] \setminus [-w_{i-1}, w_{i-1}]$). In this case,

$$P\big[\hat{f}_{w_i, w_{i-1}}^* > f_w^* + \epsilon\big] = P\big[\hat{f}_{w_i, w_{i-1}}^* \geq f_{w_i, w_{i-1}}^*\big] = 1, \quad i \neq i^*,$$

and

$$\prod_{i=1}^c P\big[\hat{f}_{w_i, w_{i-1}}^* - f_w^* > \epsilon\big] = P\big[\hat{f}_{w_{i^*}, w_{i^*-1}}^* - f_w^* > \epsilon\big] = \min_{i \in \{1, \dots, c\}} P\big[\hat{f}_{w_i, w_{i-1}}^* - f_w^* > \epsilon\big].$$

$$\square$$

From equation 2 it is clear that we need to lower bound $P[\hat{f}_{w_i, w_{i-1}}^* - f_{w_i, w_{i-1}}^* \leq \epsilon] \,\forall i \in \{1, \dots, c\}$. Since, the mathematical form of the bounds will be similar $\forall i$, let us for simplicity of notation denote the fidelities for the $i^{\text{th}}$ region by just the suffix $i$, i.e., denote $f_{w_i, w_{i-1}}^*$ by $f_i^*$ and similarly the minimum estimated fidelity and fidelities for other examples in that region. We thus now need to lower bound $P[\hat{f}_i^* - f_w^* \leq \epsilon]$ for the three different certification strategies proposed in Algorithm 2.

**Uniform Strategy:** This is the simplest strategy where we sample and query $Q$ examples uniformly in the region we want to certify. Let $U$ denote the uniform distribution over the input space in the $i^{\text{th}}$ region and let $F_i^{(\mathrm{u})}(.)$ denote the cumulative distribution function (cdf) of the fidelities induced by this uniform distribution, i.e. $F_i^{(\mathrm{u})}(v) \triangleq P_{\boldsymbol{r} \sim U}[\hat{f}_i^{(\boldsymbol{r})} \leq v]$ for some real $v$ and $\boldsymbol{r}$ belonging to the $i^{\text{th}}$ region. Then if $\hat{f}_i^{(\boldsymbol{r_1})}, ..., \hat{f}_i^{(\boldsymbol{r_Q})}$ are the fidelities of the $Q$ examples sampled by this strategy, we have

*Lemma 2 proof.*

$$
\begin{aligned}
P[\hat{f}_i^* - f_w^* \leq \epsilon] &= 1 - P[\hat{f}_i^* > f_w^* + \epsilon] \\
&= 1 - \prod_{j=1}^{Q} P[\hat{f}_i^{(\boldsymbol{r_j})} > f_w^* + \epsilon] \\
&= 1 - (1 - F_i^{(\mathrm{u})}(f_w^* + \epsilon))^Q \\
&\geq 1 - \exp\left(-Q F_i^{(\mathrm{u})}(f_w^* + \epsilon)\right)
\end{aligned}
\tag{9}
$$

$\square$

In the last step we use the following inequality for $x \in [0, 1]$, $(1 - x)^n \leq \exp -nx$.

**Uniform Incremental Strategy:** In this strategy, we sample $n \leq q$ samples uniformly $\lfloor \log(Q) \rfloor$ times. Then using each of them as centers we sample $\lfloor \frac{q}{n} \rfloor$ examples and query them. Let the cdfs induced by each of the centers through Gaussian sampling be denoted by $F_i^{\mathcal{N}_{j,k}}(.)$, where $j$ denotes the iteration number that goes up to $\lfloor \log(Q) \rfloor$ and $k$ the $k^{\text{th}}$ sampled prototype/center. With an analogous definition as before and similar steps we get,

*Lemma 3 proof.*

$$
\begin{aligned}
P[\hat{f}_i^* - f_w^* \leq \epsilon] &= 1 - \prod_{j=1}^{\lfloor \log(Q) \rfloor} \prod_{k=1}^{n=\min(2^j, q)} \left(1 - F_i^{\mathcal{N}_{j,k}}(f_w^* + \epsilon)\right)^{\lfloor \frac{q}{n} \rfloor} \\
&\geq 1 - \exp\left(-\sum_{j=1}^{\lfloor \log(Q) \rfloor} \sum_{k=1}^{n=\min(2^j, q)} \left\lfloor \frac{q}{n} \right\rfloor F_i^{\mathcal{N}_{j,k}}(f_w^* + \epsilon)\right) \\
&\geq 1 - \exp\left(-\sum_{j=1}^{\lfloor \log(Q) \rfloor} \max_{k \in \{1,...,n\}} \left\lfloor \frac{q}{n} \right\rfloor F_i^{\mathcal{N}_{j,k}}(f_w^* + \epsilon)\right) \\
&\geq 1 - \exp\left(-\max_{j \in \{1,...,\lfloor \log(Q) \rfloor\}, k \in \{1,...,n\}} \left\lfloor \frac{q}{n} \right\rfloor F_i^{\mathcal{N}_{j,k}}(f_w^* + \epsilon)\right)
\end{aligned}
\tag{10}
$$

$\square$

The above expressions convey the insight that if we find a good prototype $\boldsymbol{r_{j,k}}$ implying that $F_i^{\mathcal{N}_{j,k}}(f_w^* + \epsilon)$ is high it will lead to a higher (i.e. better) lower bound than in the uniform case. Intuitively, if we find a good prototype, exploring the region around it should be beneficial to find a good estimate of $f_i^*$.

**Adaptive Incremental Strategy:** This is possibly the most interesting strategy, where we explore *adaptively* in more promising areas of the input space unlike the other two strategies. Here too let the cdfs induced by each of the centers through Gaussian sampling be denoted by $F_i^{\mathcal{N}_{j,k}}(.)$, where $j$ denotes the iteration number that goes up to $\lfloor \log(Q) \rfloor$ and $k$ the $k^{\text{th}}$ sampled prototype/center for a given $n$. W.l.o.g. assume $F_i^{\mathcal{N}_{j,k}}(.) \leq F_i^{\mathcal{N}_{j,k+1}}(.) \; \forall j \in \{1, ..., \lfloor \log(Q) \rfloor\}, k \in \{1, ..., n-1\}$ i.e. the first prototype produces the worst estimates of the minimum fidelity, while the $n^{\text{th}}$ prototype produces the best[4].

---

[4]When sampling not always will the best cdf produce the best estimate, although it will be most likely (i.e. highest probability). We assume this probability to be 1 for each cdf based on its position in the ordering of cdfs

*Lemma 4 proof.* Now we know that for $j \in \{1, ..., \lfloor \log(Q) \rfloor\}$,

$$n = \begin{cases} 2^j & \text{if } j2^j \leq q \\ 2^l & \text{otherwise, where } l \text{ is the largest } j \text{ such that } j2^j \leq q \end{cases} \tag{11}$$

then the number of examples that will be sampled around the $k^{\text{th}}$ prototype will be,

$$\left\lfloor \frac{(2^v - 1)q}{n \log n} \right\rfloor \quad \text{where, } v = \begin{cases} 1 & \text{if } k \in \{1, ..., \lfloor \frac{n}{2} \rfloor\} \\ 2 & \text{else if } k \in \{\lceil \frac{n}{2} \rceil, ..., \lfloor \frac{3n}{4} \rfloor\} \\ \vdots & \vdots \\ \log(n) & \text{else if } k = n \end{cases} \tag{12}$$

Then we have,

$$
\begin{aligned}
P[\hat{f}_i^* - f_w^* \leq \epsilon] &= 1 - \prod_{j=1}^{\lfloor \log(Q) \rfloor} \prod_{k=1}^{n} \left( 1 - F_i^{\mathcal{N}_{j,k}}(f_w^* + \epsilon) \right)^{\left\lfloor \frac{(2^v - 1)q}{n \log n} \right\rfloor} \\
&\geq 1 - \exp\left( - \sum_{j=1}^{\lfloor \log(Q) \rfloor} \sum_{k=1}^{n} \left\lfloor \frac{(2^v - 1)q}{n \log n} \right\rfloor F_i^{\mathcal{N}_{j,k}}(f_w^* + \epsilon) \right) \\
&\geq 1 - \exp\left( - \max_{j \in \{1, ..., \lfloor \log(Q) \rfloor\}} \left\lfloor \frac{(n-1)q}{n \log n} \right\rfloor F_i^{\mathcal{N}_{j,n}}(f_w^* + \epsilon) \right) \tag{13}
\end{aligned}
$$

$\square$

*Proposition 1 proof.* For the unif strategy we query $Q$ times uniformly so its obvious that the query budget will be $Q$.

For unifI we query $n \leq q$ samples $\lfloor \log(Q) \rfloor$ times and then using them as centers query $\lfloor \frac{q}{n} \rfloor$ examples each time. Given that $q = \lfloor \frac{Q}{\log(Q)} \rfloor$, the total number of queries is thus $n \lfloor \frac{q}{n} \rfloor \lfloor \log(Q) \rfloor = n \lfloor \frac{Q}{\log(Q)n} \rfloor \lfloor \log(Q) \rfloor \leq Q$. This value however will be very close to $Q$ which is what we want.

For adaptI too one can verify that the total query budget will be close but utmost $Q$. This is because we sample and query $\lfloor \frac{q}{m \lceil \log(n) \rceil} \rfloor$ examples for $m$ prototypes $\lceil \log(n) \rceil \lfloor \log(Q) \rfloor$ times making the total query budget used equal to $\lfloor \frac{q}{m \lceil \log(n) \rceil} \rfloor m \lceil \log(n) \rceil \lfloor \log(Q) \rfloor \leq q \lfloor \log(Q) \rfloor \leq Q$. $\square$

**Proposition 2.** $F_i(f_w^* + \epsilon) = 0$ *iff all sets of inputs in region $i$ corresponding to fidelities in $[f_w^*, f_w^* + \epsilon]$ have measure zero w.r.t. the sampling densities that are non-zero everywhere in region $i$.*

*Proof.* First direction, all sets of inputs in region $i$ corresponding to values in $[f_w^*, f_w^* + \epsilon]$ having measure zero $\implies F_i(f_w^* + \epsilon) = 0$: Since all sets of inputs in region $i$ corresponding to values in $[f_w^*, f_w^* + \epsilon]$ have measure zero this would imply that the probability of getting any value in $[f_w^*, f_w^* + \epsilon]$ would also have measure zero and hence the sum of these probabilities/measures would also be zero. Second direction, one or more sets of inputs corresponding to values in $[f_w^*, f_w^* + \epsilon]$ having non-zero measure $\implies F_i(f_w^* + \epsilon) \neq 0$: If $\exists$ a set of inputs in region $i$ whose values lie in $[f_w^*, f_w^* + \epsilon]$ with non-zero measure $p$ then $F_i(f_w^* + \epsilon) \geq p > 0$, since this set will contribute a probability mass of $p$ to its corresponding value in $[f_w^*, f_w^* + \epsilon]$. $\square$

The densities are non-zero since, we sample using Uniform for unif and Gaussians for unifI and adaptI in each bounded region.

---

for clarity of exposition. One could multiply by these probabilities for posterity, but it doesn't change the nature of the bound or its interpretation.

## B  CHARACTERIZING CDFS $F_i(.)$ FOR A PIECEWISE LINEAR BLACK BOX

Several popular classes of models are piecewise linear or piecewise constant, for example neural networks with ReLU activations (Hanin & Rolnick, 2019), trees and tree ensembles, including oblique trees (Murthy et al., 1994) and model trees (Gama, 2004). We provide a partial characterization of the cdfs $F_i(.)$ for such piecewise linear black box functions $g : \mathbb{R}^d \to [0, 1]$, a linear explanation function $e_{\boldsymbol{y}} : \mathbb{R}^d \to [0, 1]$ estimated for the point $\boldsymbol{y} \in \mathbb{R}^d$, and the following fidelity function (Dhurandhar et al., 2022; 2023; Ramamurthy et al., 2020):

$$f_{\boldsymbol{y}}(\boldsymbol{x}) \triangleq 1 - |g(\boldsymbol{x}) - e_{\boldsymbol{y}}(\boldsymbol{x})|. \tag{14}$$

Assume that the black box $g$ has $t \leq p$ linear pieces within the $i^{\text{th}}$ region $R_i$. In the $s$th piece, $s = 1, \ldots, t$, $g$ can be represented as a linear function $g_s(\boldsymbol{x}) = \beta_s^T \boldsymbol{x}$, where $\beta_s \in \mathbb{R}^d$. Moreover, the $s$th piece is geometrically a polytope, which we denote as $\mathcal{P}_{i,s} \subset \mathbb{R}^d$. The explanation $e_{\boldsymbol{y}}(\boldsymbol{x}) = \alpha_{\boldsymbol{y}}^T \boldsymbol{x}$ is linear throughout. Thus within the $s$th piece, the difference $\Delta_s(\boldsymbol{x}) = g_s(\boldsymbol{x}) - e_{\boldsymbol{y}}(\boldsymbol{x})$ that determines the fidelity equation 14 is also linear, $\Delta_s(\boldsymbol{x}) = (\beta_s - \alpha_{\boldsymbol{y}})^T \boldsymbol{x}$.

Let us first consider the unif strategy where examples are sampled uniformly from $R_i$. The distribution of fidelity values is a mixture of $t$ distributions, one corresponding to each linear piece of $g$:

$$F_i(\cdot) = \sum_{s=1}^{t} \pi_s F_{i,s}(\cdot), \tag{15}$$

where $\sum_{s=1}^{t} \pi_s = 1$. In the uniform case, the probability $\pi_s$ that the $s$th piece is active is given by the ratio of volumes $\pi_s = \text{vol}(\mathcal{P}_{i,s} \cap R_i)/\text{vol}(R_i)$. The cdf $F_{i,s}$, or, equivalently, the corresponding probability density function (pdf), is largely determined by the pdf of $\Delta_s(\boldsymbol{x})$. The property of the latter pdf that is clearest to reason about is its support. The endpoints of the support can be determined by solving two linear programs, $\Delta_{s,\text{min/max}} = \min/\max_{\boldsymbol{x} \in \mathcal{P}_{i,s} \cap R_i}(\beta_s - \alpha_{\boldsymbol{y}})^T \boldsymbol{x}$. (The shape of the pdf is harder to determine; intuitively, the density at a value $\delta$ is proportional to the volume of the $\delta$-level set of $\Delta_s(\boldsymbol{x})$ intersected with the polytope, $\text{vol}(\{\boldsymbol{x} : (\beta_s - \alpha_{\boldsymbol{y}})^T \boldsymbol{x} = \delta\} \cap \mathcal{P}_{i,s} \cap R_i)$.) Given the pdf of $\Delta_s(\boldsymbol{x})$, the absolute value operation in equation 14 corresponds to folding the pdf over the vertical axis, and the $1-$ operation flips and shifts the result. Overall, we can conclude that $F_{i,s}$ is supported on an interval that is determined by $\Delta_{s,\text{min}}$ and $\Delta_{s,\text{max}}$. A larger difference vector $(\beta_s - \alpha_{\boldsymbol{y}})$ in the $s$th piece will tend to produce larger $\Delta_{s,\text{min}}, \Delta_{s,\text{max}}$ in magnitude, and hence lower fidelities. The minimum fidelity $f_i^*$ corresponds to the largest $|\Delta_{s,\text{min}}|, |\Delta_{s,\text{max}}|$ over $s$.

We now consider how the above reasoning changes for the unifI and adaptI strategies. First, instead of a single uniform distribution of examples, we have a mixture of Gaussians $\mathcal{N}_{j,k}$ indexed by iteration number $j$ and prototype $k$. Hence equation 15 is augmented with summations over $j$ and $k$, and $\pi_s$, $F_{i,s}$ gain indices to become $\pi_s^{j,k}$, $F_{i,s}^{j,k}$. Second, instead of volumes, the weight $\pi_s^{j,k}$ is given by a ratio of probabilities under each Gaussian: $\pi_s^{j,k} = P_{\mathcal{N}_{j,k}}(\mathcal{P}_{i,s} \cap R_i)/P_{\mathcal{N}_{j,k}}(R_i)$. Third, we now have multiple pdfs of $\Delta_s(\boldsymbol{x})$ to consider, one for each Gaussian $\mathcal{N}_{j,k}$, and their shapes depend on how each Gaussian weights the points in $\mathcal{P}_{i,s} \cap R_i$. What does not change however is the support $[\Delta_{s,\text{min}}, \Delta_{s,\text{max}}]$ of $\Delta_s(\boldsymbol{x})$, as this is a geometric quantity depending on the black box $g$ and explanation $e_{\boldsymbol{y}}$ but not the distribution (uniform, $\mathcal{N}_{j,k}$, or otherwise). Hence, the same statements above apply regarding the relationship between the the difference vectors $(\beta_s - \alpha_{\boldsymbol{y}})$ and the range of fidelities, mediated by $\Delta_{s,\text{min}}, \Delta_{s,\text{max}}$.

## C  MORE EFFICIENT CERTIFICATION DETAILS

**1) Lipschitz Black Box:** Let the black box function be denoted by $g(.) : \mathcal{R}^d \to \mathcal{R}$ and the explanation function by $e_{\boldsymbol{y}}(.) : \mathcal{R}^d \to \mathcal{R}$, where the subscript $\boldsymbol{y}$ denotes that the explanation function was estimated at $\boldsymbol{y} \in \mathcal{R}^d$. Let us assume the explanation function is linear i.e., $e_{\boldsymbol{y}}(\boldsymbol{x}) = \boldsymbol{\alpha}_{\boldsymbol{y}}^T \boldsymbol{x}$ (viz. in LIME and variants), where $\boldsymbol{\alpha}_{\boldsymbol{y}} \in \mathcal{R}^d$. Let the (in)fidelity function (complement of the fidelity function) for some explanation function $e_{\boldsymbol{y}}(.)$ be then defined as,

$$\bar{f}(\boldsymbol{x}) \triangleq |g(\boldsymbol{x}) - e_{\boldsymbol{y}}(\boldsymbol{x})| \tag{16}$$

Now for certification, we would want to find a rectangular region $R$ in the input space such that $\bar{f}(\boldsymbol{x}) \leq \bar{\theta} \ \forall \boldsymbol{x} \in R$, where $\bar{\theta}$ is our certification level (complement of $\theta$) and $R$ is symmetric around $\boldsymbol{x}$. Given that the black box is $l$-lipschitz we would have,

$$|g(\boldsymbol{x}) - g(\boldsymbol{y})| \leq l ||\boldsymbol{x} - \boldsymbol{y}|| \tag{17}$$

for some $l > 0$. Assume for simplicity that $g(\boldsymbol{x}) = e_{\boldsymbol{x}}(\boldsymbol{x})$, i.e., the explanation function perfectly mimics the black box if it is estimated at the same input $\boldsymbol{x}$. In other words, infidelity is zero if the estimation is at the same example. Even if we allow for some error it does not fundamentally change the results[5], but our simplifying assumption conveys the main idea more clearly in our opinion.

To certify a region $R$ around $\boldsymbol{x}$ we now want to find $\forall \boldsymbol{y} \in R, |g(\boldsymbol{y}) - e_{\boldsymbol{x}}(\boldsymbol{y})| \leq \bar{\theta}$. Upper bounding the left hand side and forcing it to be $\leq \bar{\theta}$ will give us a conservative estimate of the region around $\boldsymbol{x}$ which will be certified without having to query it. Let us thus now upper bound this quantity,

$$\begin{aligned}
|g(\boldsymbol{y}) - e_{\boldsymbol{x}}(\boldsymbol{y})| &= |e_{\boldsymbol{x}}(\boldsymbol{x}) - e_{\boldsymbol{x}}(\boldsymbol{y}) + g(\boldsymbol{y}) - e_{\boldsymbol{x}}(\boldsymbol{x})| \\
&\leq |e_{\boldsymbol{x}}(\boldsymbol{x}) - e_{\boldsymbol{x}}(\boldsymbol{y})| + |g(\boldsymbol{y}) - e_{\boldsymbol{x}}(\boldsymbol{x})| \\
&= |\boldsymbol{\alpha}_{\boldsymbol{x}}^T(\boldsymbol{x} - \boldsymbol{y})| + |g(\boldsymbol{y}) - g(\boldsymbol{x})| \\
&\leq ||\boldsymbol{\alpha}_{\boldsymbol{x}}|| \cdot ||\boldsymbol{x} - \boldsymbol{y}|| + l||\boldsymbol{x} - \boldsymbol{y}|| \\
&= ||\boldsymbol{x} - \boldsymbol{y}|| \left( ||\boldsymbol{\alpha}_{\boldsymbol{x}}|| + l \right)
\end{aligned} \tag{18}$$

The derivation mainly uses Cauchy-Schwartz inequality and that $g(.)$ is $l$-lipschitz. Therefore, we can now readily obtain a certification region $R_{\boldsymbol{x}}$ which is a hypercube around $\boldsymbol{x}$ such that

$$R_{\boldsymbol{x}} \subseteq \left\{ y, \text{ where } ||\boldsymbol{x} - \boldsymbol{y}|| \leq \frac{\bar{\theta}}{||\boldsymbol{\alpha}_{\boldsymbol{x}}|| + l} \right\} \tag{19}$$

The region $R_{\boldsymbol{x}}$ can be used to set the initial lower bounds when calling Algorithm 1, rather than the typical zero. Thus, we already would have a non-trivial region that is certified before we even make a single query for reasonable values of $\bar{\theta}$.

Interestingly, one could potentially apply this approach in an alternating fashion where once a certain region is certified by our algorithm we could try to estimate how far beyond it, again conservatively, will the infidelity not worsen below $\bar{\theta}$. However, this will have to be done more carefully as our algorithm may not have certified a region with certainty and hence errors may cascade.

**2) Piecewise Linear Black Box:** In general, knowing that the black box is piecewise linear with say $p$ pieces may not help boost our certification algorithm. However, if the fidelity is computed in a way which corresponds to the number of pieces then that can potentially be very useful. For instance, again assume the explanation function is linear $e_{\boldsymbol{y}}(\boldsymbol{x}) = \boldsymbol{\alpha}_{\boldsymbol{y}}^T \boldsymbol{x}$, and that fidelity in this case is computed as the correlation between the explanation and the corresponding linear piece $\boldsymbol{\beta}_{\boldsymbol{x}}$ in the black box function (which can be estimated) as follows: $f(\boldsymbol{x}) \triangleq \frac{|\boldsymbol{\beta}_{\boldsymbol{x}}^T \boldsymbol{\alpha}_{\boldsymbol{y}}|}{||\boldsymbol{\beta}_{\boldsymbol{x}}|| \cdot ||\boldsymbol{\alpha}_{\boldsymbol{y}}||}$. In such a case, we would know that the maximum number of fidelities we would encounter for an explanation would be utmost $p$. So at any stage in our algorithm if we encounter $p$ different values for fidelity and if all of them are $\geq \theta$, then we would know that the entire input space is certified and can stop our search.

## D  MORE INTUITION ABOUT OUR APPROACH

To better understand how the different strategies work in practice we provide short video clips unif.mp4, unifI.mp4 and adaptI.mp4 in the *videos* folder of the supplement for our three strategies. The setup is the same synthetic setup that we had in the main paper where $\theta = 0.75$, the explanation also has slope $0.75$ and the black box is a piecewise linear function with three pieces with $\boldsymbol{x}$ being at the origin. We set $d = 2$ for ease of visualization and hence the optimal half-width in this case is $0.5$. $Q$ was set to 1000.

The videos show the behavior of the methods as $z$ increases from 1 to $Z$ and the *For* loops for unifI and adaptI are rolled out showcasing the specific search patterns where the former successively samples an increasing number of prototypes (blue) but the samples around it (green circles) are reduced to meet the query budget constraint, while the latter in addition to this behavior also prunes

---

[5]Final bound is shifted proportional to the error.

half the prototypes whose minimum fidelity samples have the highest values until we reach a single prototype with a sample that has the lowest minimum fidelity value. If a sample with fidelity below 0.75 is found (i.e. violating example indicated by a red cross) then both unifI and adaptI will stop the search for that particular iteration of $z$, thus potentially using fewer than $Q$ queries. The hypercubes (i.e. squares for $d = 2$) corresponding to $lb$ and $ub$ for any $z$ are depicted by lightgreen and lightblue respectively.

# E    TOPICS RELATED TO EXTREME VALUE THEORY

**i.i.d. unifI strategy**    To facilitate the application of EVT, we use a variant of the unifI strategy that samples examples in an i.i.d. manner, as opposed to requiring a fixed number $\lfloor \frac{q}{n} \rfloor$ of samples from each Gaussian component (see Algorithm 2) which is not i.i.d. The key is to regard the examples as being drawn from a mixture distribution, specifically a mixture of Gaussian mixtures. As in Algorithm 2, the outer mixture consists of $\lfloor \log(Q) \rfloor$ Gaussian mixtures indexed by $i$, and each Gaussian mixture has $n$ components where $n = \min(2^i, q)$. Instead of drawing the same number of samples from each Gaussian mixture and each Gaussian, we use uniform mixture weights. The overall mixture distribution is therefore

$$\sum_{i=1}^{\lfloor \log(Q) \rfloor} \frac{1}{\lfloor \log(Q) \rfloor} \sum_{j=1}^{n_i = \min(2^i, q)} \frac{1}{n_i} \mathcal{N}_{i,j}(\boldsymbol{r}_j, \sigma^2 I).$$

As with the unifI strategy, we first sample the centers $\boldsymbol{r}_j$ uniformly, and then sample $Q$ examples from the above mixture distribution for querying the black-box model.

**Proof of Corollary 1**    A direct translation of the results of de Haan (1981), De Carvalho (2011, Thm. 2.3) is as follows:

$$P \left[ \hat{f}_i^* - f_i^* \leq \underbrace{\frac{\hat{\hat{f}}_i^* - \hat{f}_i^*}{(1-p)^{-1/\kappa} - 1}}_{\epsilon_i^{\text{EVT}}} \right] = 1 - p. \tag{20}$$

We then set the quantity $\epsilon_i^{\text{EVT}}$ equal to a given tolerance $\epsilon$ and solve for the corresponding value of $1 - p$. After a bit of algebra, this yields the expression in the corollary.

**Using Corollary 1 as an alternative to Theorem 1**    As discussed in the main text, Corollary 1 differs from Lemmas 2–4 in having the region-specific minimum $f_i^*$ instead of $f_w^*$, but for $i = i^* \in \arg\min_i f_i^*$, we have $f_{i^*}^* = f_w^*$. Hence

$$P \left[ \hat{f}_{i^*}^* - f_w^* \leq \epsilon \right] = P \left[ \hat{f}_{i^*}^* - f_{i^*}^* \leq \epsilon \right] = \left( 1 + \frac{\hat{\hat{f}}_{i^*}^* - \hat{f}_{i^*}^*}{\epsilon} \right)^{-\kappa}.$$

On the other hand, Lemma 1 implies that (recalling the shorthand $\hat{f}_i^* = \hat{f}_{w_i, w_{i-1}}^*$)

$$P \left[ \hat{f}_w^* - f_w^* \leq \epsilon \right] \geq \max_{i \in \{1, \dots, c\}} P \left[ \hat{f}_i^* - f_w^* \leq \epsilon \right] \geq P \left[ \hat{f}_{i^*}^* - f_w^* \leq \epsilon \right].$$

Hence

$$P \left[ \hat{f}_w^* - f_w^* \leq \epsilon \right] \geq \left( 1 + \frac{\hat{\hat{f}}_{i^*}^* - \hat{f}_{i^*}^*}{\epsilon} \right)^{-\kappa},$$

and Corollary 1 for $i = i^*$ provides a valid alternative to Theorem 1 as claimed.

**More interpretable simplification of $\epsilon_i^{\text{EVT}}$**    We now provide an upper bound on the confidence interval width $\epsilon_i^{\text{EVT}}$ in equation 20 that is simpler to interpret. Denoting this upper bound as $\hat{\epsilon}_i^{\text{EVT}}$, it follows that

$$P \left[ \hat{f}_i^* - f_i^* \leq \hat{\epsilon}_i^{\text{EVT}} \right] \geq 1 - p,$$

i.e., $[\hat{f}_i^* - \hat{\epsilon}_i^{\mathrm{EVT}}, \hat{f}_i^*]$ is also (at least) a $(1-p)$-confidence interval for the minimum fidelity $f_i^*$.

To bound $\epsilon_i^{\mathrm{EVT}}$ from above, it is equivalent to bounding the denominator $((1-p)^{-1/\kappa} - 1)$ from below since the numerator $\hat{\hat{f}}_i^* - \hat{f}_i^*$ is non-negative. We regard the denominator as a function of $1/\kappa$, $D(1/\kappa) = ((1-p)^{-1/\kappa} - 1)$. This is an exponential function and hence convex in $1/\kappa$. It is therefore bounded from below by its tangent line at $1/\kappa = 0$:

$$D\left(\frac{1}{\kappa}\right) \geq D(0) + \frac{D'(0)}{\kappa} = 0 - \frac{\log(1-p)}{\kappa} = \frac{\log(1/(1-p))}{\kappa}.$$

Hence

$$\epsilon_i^{\mathrm{EVT}} \leq \hat{\epsilon}_i^{\mathrm{EVT}} = \frac{\kappa(\hat{\hat{f}}_i^* - \hat{f}_i^*)}{\log(1/(1-p))}. \tag{21}$$

The upper bound in equation 21 is proportional to parameter $\kappa$ of the extreme value distribution and to the difference $(\hat{\hat{f}}_i^* - \hat{f}_i^*)$ between the smallest and second-smallest observed fidelities. It also depends logarithmically on the confidence level $1-p$. As noted in Section 6.2, $\kappa$ is often taken to be $d/2$ (de Haan, 1981) (recall that $d$ is the input dimension). In this case, $\hat{\epsilon}_i^{\mathrm{EVT}}$ is proportional to the dimension. The upper bound becomes tighter as $1/\kappa \to 0$, i.e., in the limit of high $\kappa$ and high dimension.

Table 2: Synthetic results for $x = [0]^d$, $Z = 10$, $\theta = 0.75$, explanation is slope $0.75$ hyperplane and optimal half-width is $\frac{1}{d}$ with standard errors (rounded to 3 decimal places, where if value is 0 after rounding we do not state it).

| $d$ | $Q$ | unif | | unifI | | adaptI | | $\mathrm{ZO}^+$ | |
|---|---|---|---|---|---|---|---|---|---|
| | | $w$ | Time (s) | $w$ | Time (s) | $w$ | Time (s) | $w$ | Time (s) |
| 1 | 10 | 1 | .001 | 1 | .001 | 1 | .001 | 1 | .012 |
| | $10^2$ | 1 | .006 | 1 | .004 | 1 | .002 | 1 | 1.221 |
| | $10^3$ | 1 | .055 | 1 | .041 | 1 | .026 | 1 | 1.724 |
| | $10^4$ | 1 | .53 | 1 | .418 | 1 | .189 | 1 | 1.641 |
| 10 | 10 | $.06 \pm .027$ | .001 | $.037 \pm .028$ | .001 | $.142 \pm .12$ | .001 | $.3 \pm .07$ | .012 |
| | $10^2$ | $.082 \pm .019$ | .003 | $.06 \pm .023$ | .007 | $.08 \pm .02$ | .003 | .1 | .125 |
| | $10^3$ | $.09 \pm .018$ | .036 | $.085 \pm .019$ | .049 | $.11 \pm .02$ | .044 | .1 | 1.354 |
| | $10^4$ | $.1 \pm .016$ | .363 | $.117 \pm .008$ | .615 | $.1 \pm .01$ | .551 | .1 | 14.944 |
| $10^2$ | 10 | $.012 \pm .005$ | .001 | $.006 \pm .002$ | .001 | $.007 \pm .09$ | .001 | .05 | .031 |
| | $10^2$ | $.012 \pm .003$ | .005 | $.007 \pm .002$ | .012 | $.008 \pm .002$ | .005 | .025 | .3 |
| | $10^3$ | $.011 \pm .004$ | .054 | $.009 \pm .001$ | .158 | $.01 \pm .001$ | .09 | .012 | 4.072 |
| | $10^4$ | $.01 \pm .003$ | .632 | $.01 \pm .001$ | 1.692 | $.01 \pm .001$ | .51 | .009 | 55.87 |
| $10^3$ | 10 | $5 \times 10^{-4}$ | .003 | $3 \times 10^{-4}$ | .004 | $5 \times 10^{-4}$ | .002 | $.037 \pm .008$ | .307 |
| | $10^2$ | $6 \times 10^{-4}$ | .011 | .001 | .073 | $6 \times 10^{-4}$ | .044 | .012 | 2.579 |
| | $10^3$ | $8 \times 10^{-4}$ | .077 | .001 | 1.074 | $8 \times 10^{-4}$ | .511 | .003 | 28.335 |
| | $10^4$ | .001 | .588 | .001 | 13.786 | $9 \times 10^{-4}$ | 5.097 | .001 | 288.523 |
| $10^4$ | 10 | $6.3 \times 10^{-5}$ | .012 | $5.1 \times 10^{-5}$ | .098 | $5.8 \times 10^{-5}$ | .021 | $.006 \pm .001$ | 3.76 |
| | $10^2$ | $6.6 \times 10^{-5}$ | .072 | $7.7 \times 10^{-5}$ | 1.187 | $7.8 \times 10^{-5}$ | .43 | .004 | 34.602 |
| | $10^3$ | $8.3 \times 10^{-5}$ | .771 | $8.4 \times 10^{-5}$ | 12.452 | $8.5 \times 10^{-5}$ | 7.91 | $8.4 \times 10^{-4}$ | 391.494 |
| | $10^4$ | $8.9 \times 10^{-5}$ | 4.83 | $9.1 \times 10^{-5}$ | 112.58 | $9.4 \times 10^{-5}$ | 88.342 | $9.3 \times 10^{-5}$ | 4384.76 |

## F  EXPERIMENTAL DETAILS AND MORE RESULTS

In our implementation of Ecertify, we also have an additional exit condition that checks how close $lb$ and $ub$ are in any iteration of $Z$. If the difference is less than $\frac{0.1}{d}$, we return the current best solution. This prevents the strategies from trying to find samples in this very narrow (low volume) region, which can be difficult.

**Choices for the upper bound $B$:**   In Section 4, we introduced the variable $B$ (from Algorithm 1) that bounds the half-widths to consider from above. Recall that in Algorithm 1, once a violator ($\boldsymbol{b}$) is found, we shrink the candidate region by setting the new $ub$ to be the midpoint between $B$ and $lb$ (the width of the last certified region), and $B$ was set to be the *minimum* of all the coordinates of $\boldsymbol{b}$ that exceed $lb$. In this experiment, we consider other choices for setting $B$: i) to the *maximum* value and ii) to the *mean* value of the coordinates of $\boldsymbol{b}$ that exceed $lb$.

Table 3: Synthetic results for $x = [0]^d$, $Z = 10$, $\theta = 0.9$, explanation is slope 0.9 hyperplane and optimal half-width is $\frac{1}{d}$ with standard errors (rounded to 3 decimal places, where if value is 0 after rounding we do not state it).

| $d$ | $Q$ | unif | | unifI | | adaptI | | ZO$^+$ | |
|---|---|---|---|---|---|---|---|---|---|
| | | $w$ | Time (s) | $w$ | Time (s) | $w$ | Time (s) | $w$ | Time (s) |
| 1 | 10 | 1 | .001 | 1 | .001 | 1 | .001 | 1 | .011 |
| | $10^2$ | 1 | .004 | 1 | .005 | 1 | .002 | 1 | 1.122 |
| | $10^3$ | 1 | .06 | 1 | .04 | 1 | .024 | 1 | 1.678 |
| | $10^4$ | 1 | .55 | 1 | .42 | 1 | .181 | 1 | 1.666 |
| 10 | 10 | $.071 \pm .022$ | .001 | $.039 \pm .027$ | .001 | $.133 \pm .11$ | .001 | $.2 \pm .09$ | .015 |
| | $10^2$ | $.083 \pm .017$ | .002 | $.063 \pm .019$ | .005 | $.08 \pm .02$ | .003 | .1 | .128 |
| | $10^3$ | $.11 \pm .012$ | .032 | $.081 \pm .021$ | .051 | $.11 \pm .01$ | .048 | .1 | 1.411 |
| | $10^4$ | $.1 \pm .011$ | .369 | $.115 \pm .009$ | .623 | $.1 \pm .008$ | .573 | .1 | 14.871 |
| $10^2$ | 10 | $.013 \pm .004$ | .001 | $.006 \pm .002$ | .001 | $.008 \pm .08$ | .001 | .053 | .034 |
| | $10^2$ | $.013 \pm .003$ | .005 | $.006 \pm .003$ | .014 | $.009 \pm .001$ | .004 | .027 | .35 |
| | $10^3$ | $.01 \pm .005$ | .051 | $.01 \pm .002$ | .162 | $.012 \pm .001$ | .11 | .011 | 4.113 |
| | $10^4$ | $.01 \pm .002$ | .655 | $.01 \pm .001$ | 1.679 | .01 | .56 | .01 | 58.12 |
| $10^3$ | 10 | $5.3 \times 10^{-4}$ | .003 | $5.7 \times 10^{-4}$ | .004 | $5.9 \times 10^{-4}$ | .002 | $.028 \pm .01$ | .317 |
| | $10^2$ | $6.3 \times 10^{-4}$ | .014 | $9.2 \times 10^{-4}$ | .077 | $7.9 \times 10^{-4}$ | .042 | .009 | 2.636 |
| | $10^3$ | $8.5 \times 10^{-4}$ | .079 | .001 | 1.051 | $9.1 \times 10^{-4}$ | .563 | .002 | 29.638 |
| | $10^4$ | .001 | .613 | .001 | 12.673 | $9.9 \times 10^{-4}$ | 5.165 | .001 | 291.122 |
| $10^4$ | 10 | $6.7 \times 10^{-5}$ | .011 | $6.1 \times 10^{-5}$ | .123 | $6.6 \times 10^{-5}$ | .03 | $.004 \pm .001$ | 3.83 |
| | $10^2$ | $7.4 \times 10^{-5}$ | .074 | $7.9 \times 10^{-5}$ | 1.221 | $8.2 \times 10^{-5}$ | .48 | .002 | 36.671 |
| | $10^3$ | $8.7 \times 10^{-5}$ | .777 | $8.9 \times 10^{-5}$ | 12.53 | $9.3 \times 10^{-5}$ | 8.16 | $8.6 \times 10^{-4}$ | 401.821 |
| | $10^4$ | $9.5 \times 10^{-5}$ | 5.01 | $9.6 \times 10^{-5}$ | 101.99 | $9.9 \times 10^{-5}$ | 90.112 | $9.6 \times 10^{-5}$ | 4517.119 |

Table 4: Synthetic results for $x = [0]^d$, $Z = 10$, $\theta = 0.5$, explanation is slope 0.5 hyperplane and optimal half-width is $\frac{1}{d}$ with standard errors (rounded to 3 decimal places, where if value is 0 after rounding we do not state it).

| $d$ | $Q$ | unif | | unifI | | adaptI | | ZO$^+$ | |
|---|---|---|---|---|---|---|---|---|---|
| | | $w$ | Time (s) | $w$ | Time (s) | $w$ | Time (s) | $w$ | Time (s) |
| 1 | 10 | 1 | .001 | 1 | .001 | 1 | .001 | 1 | .015 |
| | $10^2$ | 1 | .005 | 1 | .005 | 1 | .004 | 1 | 1.312 |
| | $10^3$ | 1 | .051 | 1 | .039 | 1 | .023 | 1 | 1.756 |
| | $10^4$ | 1 | .51 | 1 | .432 | 1 | .181 | 1 | 1.611 |
| 10 | 10 | $.07 \pm .028$ | .001 | $.032 \pm .025$ | .001 | $.122 \pm .13$ | .001 | $.24 \pm .08$ | .011 |
| | $10^2$ | $.078 \pm .012$ | .004 | $.061 \pm .022$ | .005 | $.08 \pm .02$ | .003 | .1 | .133 |
| | $10^3$ | $.09 \pm .01$ | .04 | $.083 \pm .015$ | .047 | $.12 \pm .01$ | .046 | .1 | 1.211 |
| | $10^4$ | $.1 \pm .013$ | .351 | $.109 \pm .005$ | .622 | $.1 \pm .05$ | .566 | .1 | 15.175 |
| $10^2$ | 10 | $.012 \pm .006$ | .001 | $.007 \pm .001$ | .001 | $.008 \pm .08$ | .001 | .06 | .037 |
| | $10^2$ | $.011 \pm .003$ | .006 | $.008 \pm .002$ | .014 | $.008 \pm .002$ | .006 | .023 | .312 |
| | $10^3$ | $.011 \pm .002$ | .051 | $.009 \pm .001$ | .151 | $.012 \pm .001$ | .10 | .013 | 4.178 |
| | $10^4$ | $.01 \pm .002$ | .611 | $.01 \pm .002$ | 1.633 | $.01 \pm .001$ | .46 | .01 | 58.62 |
| $10^3$ | 10 | $5.2 \times 10^{-4}$ | .004 | $4.8 \times 10^{-4}$ | .005 | $5.3 \times 10^{-4}$ | .002 | $.023 \pm .008$ | .365 |
| | $10^2$ | $6 \times 10^{-4}$ | .016 | $8 \times 10^{-4}$ | .075 | $7.3 \times 10^{-4}$ | .046 | .009 | 2.677 |
| | $10^3$ | $8.1 \times 10^{-4}$ | .075 | .001 | 1.033 | $8.8 \times 10^{-4}$ | .525 | .003 | 27.547 |
| | $10^4$ | .001 | .595 | .001 | 12.976 | $9.9 \times 10^{-4}$ | 4.98 | .001 | 285.479 |
| $10^4$ | 10 | $6.4 \times 10^{-5}$ | .01 | $6.1 \times 10^{-5}$ | .101 | $6.8 \times 10^{-5}$ | .019 | $.003 \pm .002$ | 3.82 |
| | $10^2$ | $6.7 \times 10^{-5}$ | .073 | $7.9 \times 10^{-5}$ | 1.234 | $8.1 \times 10^{-5}$ | .46 | .002 | 33.1 |
| | $10^3$ | $8.2 \times 10^{-5}$ | .775 | $8.5 \times 10^{-5}$ | 12.437 | $8.9 \times 10^{-5}$ | 8.12 | $8.5 \times 10^{-4}$ | 389.352 |
| | $10^4$ | $9.1 \times 10^{-5}$ | 4.78 | $9.4 \times 10^{-5}$ | 115.01 | $9.7 \times 10^{-5}$ | 90.103 | $9.4 \times 10^{-5}$ | 4291.438 |

We choose the synthetic data set-up as described in Section 7 for this experiment as the true certified half-widths are known ($1/d$). In Figures 3 and 4, we report the found $w$'s and timings for the three variations of $B$ (for each of the three strategies).

In Figure 3, we observe that for smaller dimensions ($\approx$ 10s) the choice of $B$ has negligible effect, but for higher dimensions taking the *minimum* provides much more accurate estimates of the true half-width albeit slightly conservative, while both *max* and *mean* choices overestimate the true half-width. The reason for this is as follows: note that once the upper bound $B$ is set, the resulting certified half-width $w$ could at best converge to $B$, and thus setting $B$ to be the maximum (or mean) of the violator ($\boldsymbol{b}$)'s coordinates can be overly optimistic.

In Figure 4, we observe *min* also enjoys the benefit of faster running time, since it brings about the largest reduction in (candidate) widths to consider. This analysis supports our choice of using *min* in Algorithm 1 as well as in our implementation.

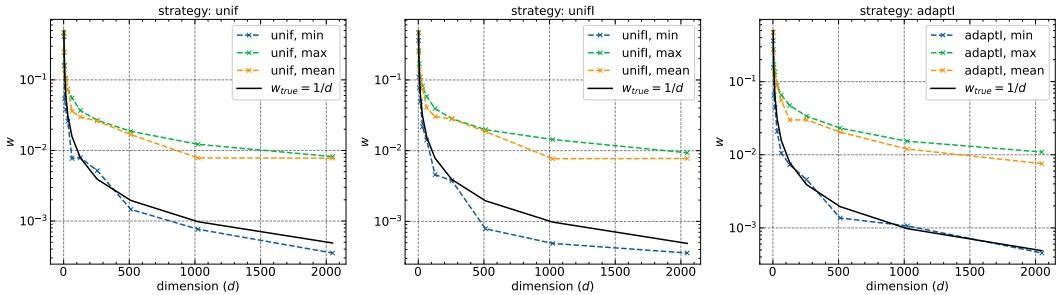

Figure 3: Certified half-widths vs. dimensions plots for the synthetic data set-up with different choices of setting $B$ for the proposed 3 strategies. Note that, choosing *min* is (slightly) conservative as it is almost always below the true certified width (the black solid curve) and both *max* and *mean* overestimate the true width (the y-axis is in log-scale).

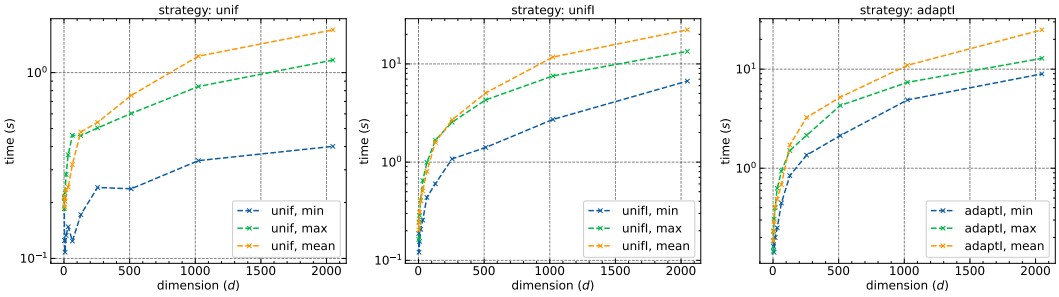

Figure 4: Timing charts for the 3 strategies with different choices for $B$. We see that choosing *min* results in shorter run-times for all three strategies (the y-axis is in log-scale) since it brings about the largest reduction in candidate spatial widths to consider.

**LIME setting:** For tabular classification datasets (heloc and arrhythmia), we obtained LIME explanations by using 1000 samples around each instance, and with top 5 features in the explanation. We did not discretize continuous features. We constructed the linear explanation using the coefficients and intercepts from the explanation to apply it to other instances.

For images applying LIME explanations is not straightforward since each image has its own mask based on the superpixels it identifies. Hence to apply explanations across images we identified coefficient values for each pixel in the input image and then depending on the (absolute value of the) mask for a sampled image in the current to be certified region summed the relevant coefficients. This intuitively is equivalent multiplying coefficients with feature values for an example. We also reduced the neighbourhood sample size for LIME to 100 (for CIFAR10) and to 10 (for ImageNet) as this mask finding procedure was time consuming, especially when the certification was done with a high query budget.

**SHAP setting:** For SHAP, we used the model agnostic KernelSHAP explainer and used mean values of features from training data as the background values. We obtained shapley values for an instance by taking 10 features and 1000 explanation samples (`nsamples`) for tabular, and 1000 features and 500 `nsamples` for image datasets. To apply the obtained SHAP explanation on other examples, we obtained an equivalent linear regression model from the shapley values following Amparore et al. (2021).

### F.1 ADDITIONAL SYNTHETIC EXPERIMENTS

As mentioned in the main paper we report results here on more cases along with their standard errors. In Table 2 we see results of Table 1 with standard errors. Tables 3 and 4 report results for $\theta = 0.9$ and $\theta = 0.5$ respectively where the explanations are hyperplanes having a $\theta$ slope with all the axes. As we can see the insights discussed in the main paper carry over for these cases too.

### F.2 ADDITIONAL REAL EXPERIMENTS

In Figures 5, 6, 7 and 8 we see qualitatively similar behavior[6] as discussed in the main paper, where in terms of half-widths, in general, unif seems to be the best for the lower dimensional datasets such as HELOC and Arrhythmia, while unifI is best for CIFAR10 and adaptI is best for ImageNet. Again adaptI seems to be the fastest, possibly because of the high efficiency in rejection sampling and it honing on to the violating examples in a region with (much) fewer queries than the allotted budget $Q$ on average.

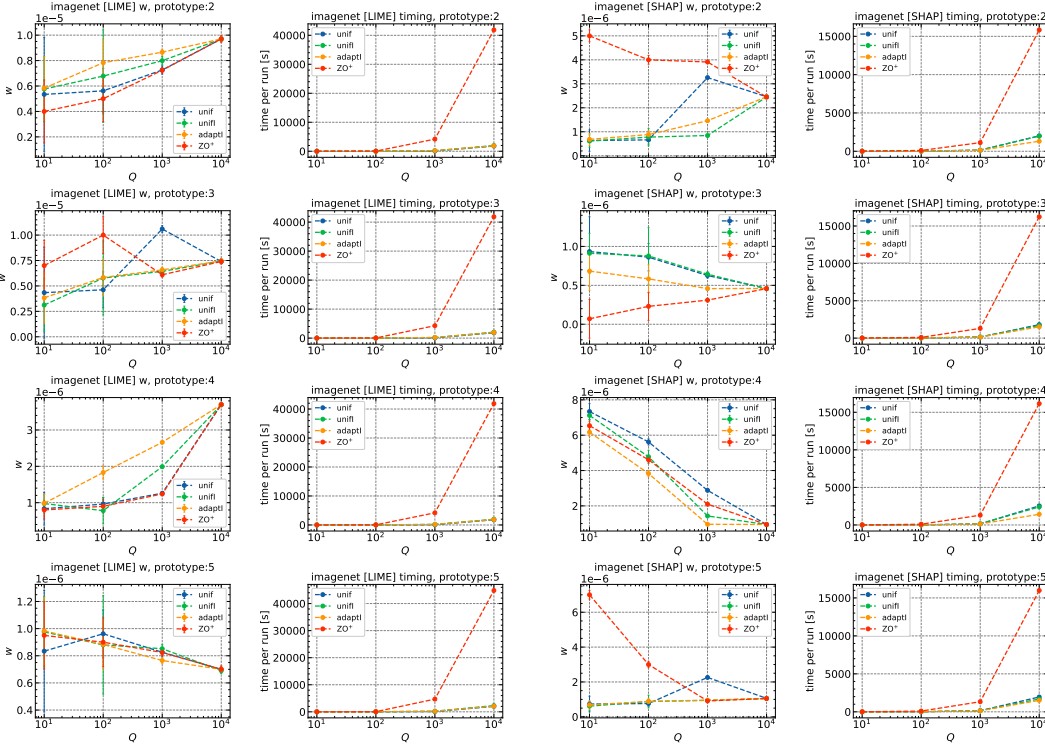

Figure 5: Rows 1-4 above correspond to prototypes 2-5 from the ImageNet dataset. The first prototype results are in the main paper. First two columns are LIME half-width and timing results, while last two columns are SHAP half-width and timing results respectively.

---

[6]In Figure 6 $ZO^+$ exited without returning a half-width for $Q = 10$ on CIFAR10 and hence it is not plotted for that value. For prototype 2 using LIME $ZO^+$ again exited immediately for $Q = 10000$ and hence the zero values for time and half-width.

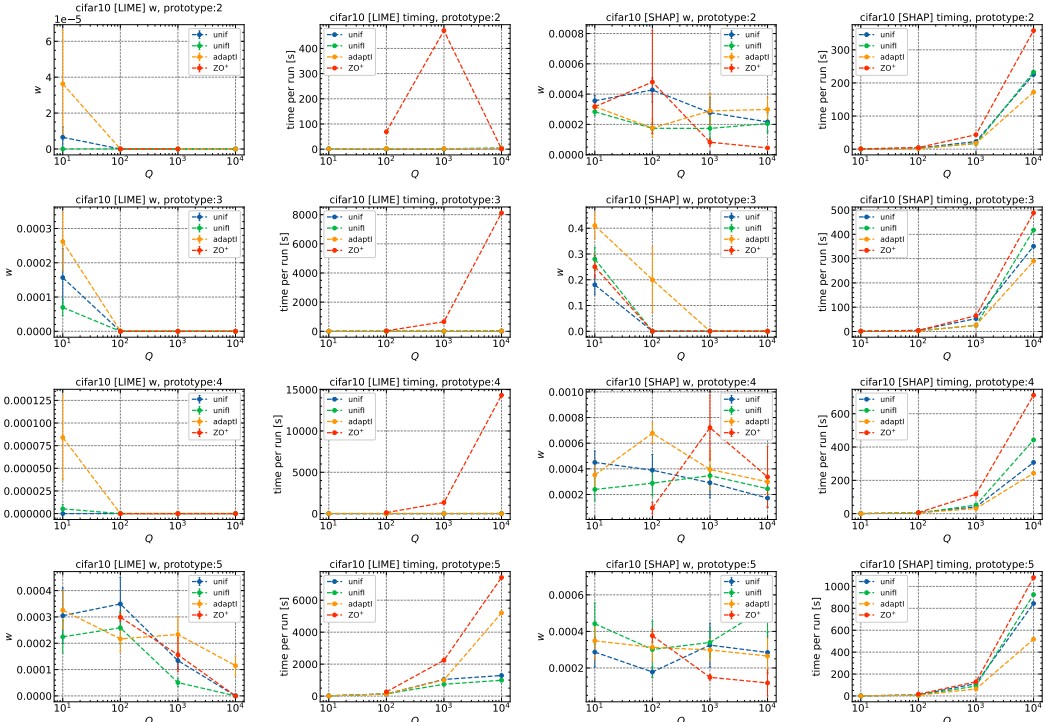

Figure 6: Rows 1-4 above correspond to prototypes 2-5 from the CIFAR10 dataset. The first prototype results are in the main paper. First two columns are LIME half-width and timing results, while last two columns are SHAP half-width and timing results respectively.

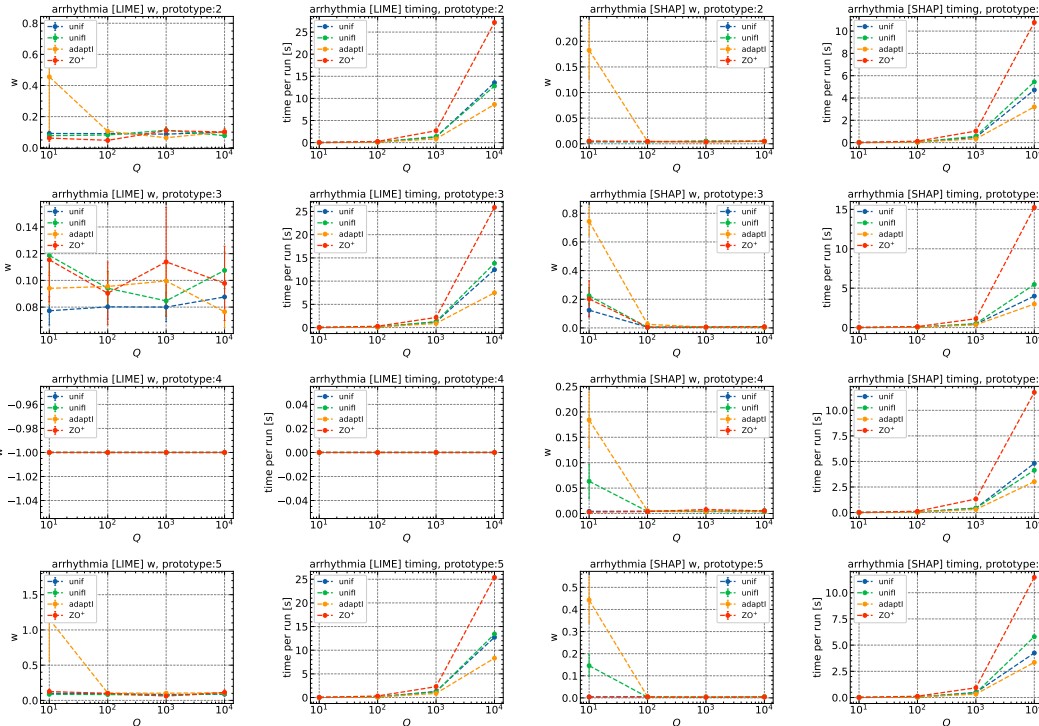

Figure 7: Rows 1-4 above correspond to prototypes 2-5 from the arrhythmia dataset. The first prototype results are in the main paper. First two columns are LIME half-width and timing results, while last two columns are SHAP half-width and timing results respectively.

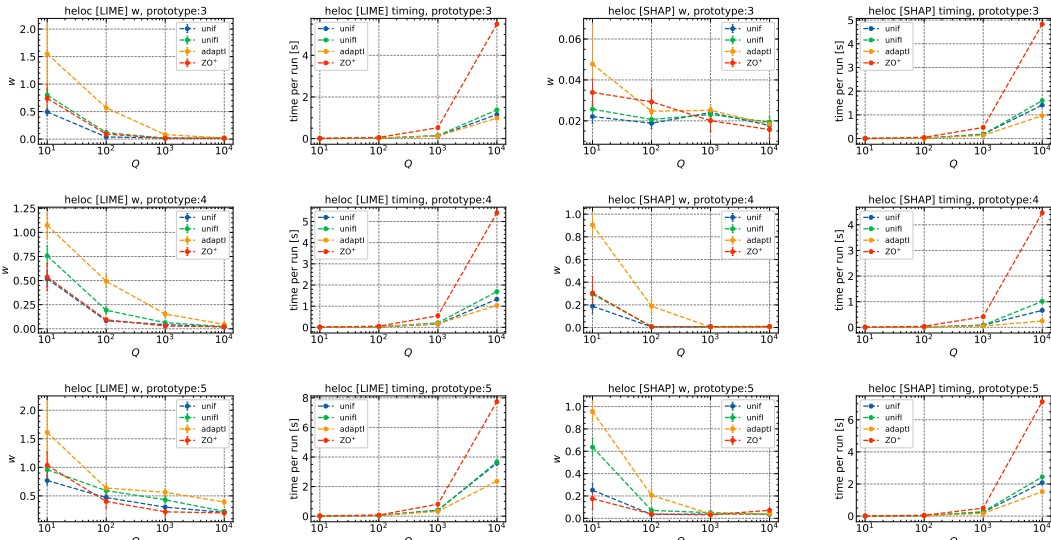

Figure 8: Rows 1-3 above correspond to prototypes 3-5 from the HELOC dataset. The first two prototype results are in the main paper. First two columns are LIME half-width and timing results, while last two columns are SHAP half-width and timing results respectively.

## G    EXPANDED DISCUSSION SECTION

Rather than certified hypercubes, one could also find hyper-rectangles or even arbitrary $\ell_p$ balls (not just $\ell_\infty$) with our strategies for which too the main theoretical results should apply. This would nonetheless require extra book-keeping to correctly demarcate the certified (and violating) boundaries in each case. From a practical standpoint the strategies could also return the nearest violating example (i.e. fidelity $< \theta$) than the minimum fidelity one reducing the search space even faster. Moreover, the outer *For* loop in unifI and adaptI can be parallelized.

There are multiple interesting future directions which we discuss below.

**Applicability to other explanation method types:**    Although our experiments considered feature based explanations, one could apply our approach even to contrastive (Dhurandhar et al., 2018; Wachter et al., 2017) or exemplar (Kim et al., 2016; Gurumoorthy et al., 2019) based explanations, as long as one can apply the given explanation (e.g. a class-changing perturbation) to different examples and measure the resulting quality. For instance, in the case of contrastive explanations, one could apply the same perturbations that change the class of one input example to other examples and check if the same class change occurs. This would be a potential quality metric in this case.

**Multi-armed bandits and hyperparameter optimization:** Multi-armed bandit algorithms (Slivkins, 2019) could possibly be adapted to our setting, especially those designed for infinitely many arms (Auer et al., 2003). We note however that they typically make assumptions such as Lipschitzness, local smoothness or bounded near-optimality dimension (Bubeck et al., 2011), which we did not have to make for our main results.

It is worth mentioning that our certification strategies have relations to methods employed in hyperparameter optimization (Tong Yu, 2020), where efficient search of the hyperparameter space is needed. However, in addition to the domain being completely different, methods in hyperparameter optimization try to find hyperparameter values that will result in the best performing model w.r.t. a certain quality metric such as accuracy. In our case, we do not have an already provided model that we wish to optimize and for which we have to assign computational resources to train. Rather, we have to decipher an intelligent way to find low fidelity examples in a compact region of the input space. It is a priori unclear how the query budget can be effectively assigned and used (viz. sampling prototypes, perturbing them, etc.) in such a setup. Moreover, theoretical results in their domain typically involve making additional assumptions about the loss behavior with more training, something that wasn't required for us to prove the bounds, not to mention them having little relevance in our setup.

**Known upper bound:** If a priori we know an upper bound on the certification region, there could be more intelligent ways of assigning the query budget to Certify() in Algorithm 1, rather than simply a fixed budget of $Q$. One could possibly keep querying in the region until the certification criterion is violated. If a violation is found then the new region to certify would be the hypercube contained within the closest dimension of this example to the input. Now repeat the process with the remaining query budget in this new region. Once we exhaust the query budget declare the current region being certified as a valid certified region.

**Applicability to manifolds:** It would be desirable to adapt these methods to work on lower dimensional manifolds. As a first step, one could simply apply the current methods to the latent space (e.g. as learned by an auto-encoder) rather than the input space. Thus, although the regions will be hypercubes in the latent space they will be more free-form in the input space which might be interesting.

**Designing new explanation methods:** One could design new explanation methods that maximize the size of the certification region while also being faithful. Ideas from constraint generation (Dash et al., 2018) could be used here where the identified violating examples would serve as the constraints that get added when finding a suitable explanation possibly leading to more robust explanations.

**Limitations:** Our approaches rely on random sampling and have probabilistic guarantees, hence in any particular case it is possible that the half-widths reported may be different than the true half-widths. Also results may vary run to run. Possible mitigation is by averaging over multiple runs and/or using sufficient query budget.

## H BOUND COMPUTATION

In Tables 5 and 7 we see the bounds computed using Theorem 1 for the synthetic and real data experiments.

Table 5: Bounds computed based on Theorem 1 for the synthetic results reported in Table 1. $\epsilon$ was set to 0.01 and kernel density estimation (kde) using the scipy library with default settings was done to estimate the distribution of fidelities. The cdfs were computed based using the true $f^*$ in the region, as well as two proxies for $f^*$, namely $\hat{f}^*$ i.e. minimum estimated fidelity for the respective region based on the fidelities returned by the algorithms, and $\theta$ i.e. fidelity threshold passed as input to the algorithms. Also time to compute the bounds is reported given that we have the fidelities already available for samples from running the respective strategies. The time is an average over three options for $f^*$ mentioned above each of which takes similar time. As can be seen the bounds using $f^*$ or its estimate $\hat{f}^*$ are quite close, while the bounds using $\theta$ as a proxy for $f^*$ are slightly conservative. We can see that adaptI converges to probability 1 the fastest in terms of $Q$, probably because of its ability to hone in on the low fidelity regions leading to higher values of the cdf and hence tighter bounds. In terms of time, unif is the fastest since we just have to estimate a single cdf for each region we certify. For adaptI, approximately $\log(Q)$ cdfs have to be estimated per region, while for unifI it is $\approx q(1 + \log\log(Q))$, which leads to the higher time. Nonetheless, in practical terms, all bounds seem to be reasonably efficient to compute (within $\sim 6$ minutes here).

| $d$ | $Q$ | unif | | | | unifI | | | | adaptI | | | |
|---|---|---|---|---|---|---|---|---|---|---|---|---|---|
| | | $f^*$ | $\hat{f}^*$ | $\theta$ | Time (s) | $f^*$ | $\hat{f}^*$ | $\theta$ | Time (s) | $f^*$ | $\hat{f}^*$ | $\theta$ | Time (s) |
| 1 | 10 | 0.001 | 0.001 | 0.001 | 0.123 | 0.001 | 0.001 | 0.001 | 0.127 | 0.001 | 0.001 | 0.001 | 0.135 |
| | $10^2$ | 0.457 | 0.457 | 0.457 | 0.158 | 0.764 | 0.764 | 0.764 | 1.312 | 0.971 | 0.971 | 0.971 | 0.411 |
| | $10^3$ | 1.000 | 1.000 | 1.000 | 0.213 | 1.000 | 1.000 | 1.000 | 12.623 | 1.000 | 1.000 | 1.000 | 0.801 |
| | $10^4$ | 1.000 | 1.000 | 1.000 | 0.727 | 1.000 | 1.000 | 1.000 | 368.671 | 1.000 | 1.000 | 1.000 | 3.638 |
| 10 | 10 | 0.001 | 0.001 | 0.001 | 0.125 | 0.001 | 0.001 | 0.001 | 0.129 | 0.001 | 0.001 | 0.001 | 0.132 |
| | $10^2$ | 0.461 | 0.461 | 0.445 | 0.162 | 0.764 | 0.764 | 0.691 | 1.396 | 0.962 | 0.962 | 0.949 | 0.405 |
| | $10^3$ | 1.000 | 1.000 | 1.000 | 0.217 | 1.000 | 1.000 | 1.000 | 13.142 | 1.000 | 1.000 | 1.000 | 0.793 |
| | $10^4$ | 1.000 | 1.000 | 1.000 | 0.79 | 1.000 | 1.000 | 1.000 | 370.263 | 1.000 | 1.000 | 1.000 | 3.792 |
| $10^2$ | 10 | 0.001 | 0.001 | 0.001 | 0.127 | 0.001 | 0.001 | 0.001 | 0.127 | 0.001 | 0.001 | 0.001 | 0.130 |
| | $10^2$ | 0.466 | 0.468 | 0.461 | 0.161 | 0.771 | 0.773 | 0.768 | 1.387 | 0.972 | 0.971 | 0.967 | 0.431 |
| | $10^3$ | 1.000 | 1.000 | 1.000 | 0.209 | 1.000 | 1.000 | 1.000 | 13.128 | 1.000 | 1.000 | 1.000 | 0.828 |
| | $10^4$ | 1.000 | 1.000 | 1.000 | 0.789 | 1.000 | 1.000 | 1.000 | 370.527 | 1.000 | 1.000 | 1.000 | 3.716 |
| $10^3$ | 10 | 0.001 | 0.001 | 0.000 | 0.129 | 0.001 | 0.001 | 0.001 | 0.127 | 0.001 | 0.001 | 0.000 | 0.137 |
| | $10^2$ | 0.455 | 0.455 | 0.441 | 0.163 | 0.778 | 0.780 | 0.772 | 1.411 | 0.970 | 0.971 | 0.965 | 0.429 |
| | $10^3$ | 1.000 | 1.000 | 0.981 | 0.222 | 1.000 | 1.000 | 1.000 | 13.172 | 1.000 | 1.000 | 0.995 | 0.811 |
| | $10^4$ | 1.000 | 1.000 | 1.000 | 0.733 | 1.000 | 1.000 | 1.000 | 371.321 | 1.000 | 1.000 | 1.000 | 3.712 |
| $10^4$ | 10 | 0.001 | 0.001 | 0.000 | 0.130 | 0.001 | 0.001 | 0.000 | 0.125 | 0.001 | 0.001 | 0.000 | 0.140 |
| | $10^2$ | 0.449 | 0.450 | 0.444 | 0.153 | 0.765 | 0.765 | 0.759 | 1.344 | 0.972 | 0.972 | 0.969 | 0.401 |
| | $10^3$ | 1.000 | 1.000 | 0.998 | 0.235 | 1.000 | 1.000 | 0.999 | 12.763 | 1.000 | 1.000 | 1.000 | 0.789 |
| | $10^4$ | 1.000 | 1.000 | 1.000 | 0.766 | 1.000 | 1.000 | 1.000 | 373.891 | 1.000 | 1.000 | 1.000 | 3.601 |

Table 6: Probability lower bounds from EVT for the synthetic results reported in Table 1 (same as Table 5). As discussed in Section 6 and Appendix E, the bounds apply to the unif and i.i.d. unifI strategies and are based on Corollary 1. Ideally, one should apply Corollary 1 with $i = i^* \in \arg\min_i f_i^*$, but since $i^*$ is not known to the algorithms, we use $\hat{i} \in \arg\min_i \hat{f}_i^*$ as an approximation. As in Table 5, $\epsilon = 0.01$, and the exponent $\kappa$ in Corollary 1 is set to $d/2$. We make the following observations: 1) The bounds for i.i.d. unifI in particular are high enough to be meaningful. 2) At the same time, the bounds in Table 6 are weaker than those in Table 5. This appears to be the price of using an easily computable asymptotic expression rather than estimating cdfs. While the $Q = 10$ results in Table 6 might appear to be better, we recall that EVT holds in the limit of large $Q$ so the $Q = 10$ values are questionable. We include them for completeness to match Table 5. 3) The bounds in Table 6 do suffer somewhat from increasing dimension $d$, due to the exponent $\kappa = d/2$. 4) The bounds for i.i.d. unifI are much better than for unif. This supports the intuition that if one of the prototypes from unifI happens to be good (having close to minimum fidelity), then sampling more densely around it is better than sampling uniformly throughout.

| $d$ | $Q$ | unif | i.i.d. unifI |
|-----|-----|------|--------------|
| 1 | 10 | 0.624 | 0.654 |
| | $10^2$ | 0.9 | 0.875 |
| | $10^3$ | 0.989 | 0.981 |
| | $10^4$ | 0.998 | 0.998 |
| 10 | 10 | 0.077 | 0.575 |
| | $10^2$ | 0.191 | 0.553 |
| | $10^3$ | 0.244 | 0.563 |
| | $10^4$ | 0.317 | 0.645 |
| $10^2$ | 10 | 0.066 | 0.451 |
| | $10^2$ | 0.081 | 0.513 |
| | $10^3$ | 0.081 | 0.493 |
| | $10^4$ | 0.083 | 0.558 |
| $10^3$ | 10 | 0.016 | 0.416 |
| | $10^2$ | 0.081 | 0.471 |
| | $10^3$ | 0.141 | 0.479 |
| | $10^4$ | 0.109 | 0.484 |
| $10^4$ | 10 | 0.02 | 0.364 |
| | $10^2$ | 0.05 | 0.436 |
| | $10^3$ | 0.05 | 0.454 |
| | $10^4$ | 0.07 | 0.515 |

Table 7: Below we see the (estimated) lower bounds on the probability in Theorem 1 and the additional time to compute them for the example in the main paper on ImageNet (Figure 1 first row), given that we have the fidelities already available for samples from running the respective strategies. $\epsilon$ was set to $0.01$ and kernel density estimation (kde) using the scipy library with default settings was done to estimate the distribution of fidelities. The cdfs were computed based on two proxies for $f^*$ (which is unknown): i) $\hat{f}^*$ i.e. minimum estimated fidelity for the respective region based on the fidelities returned by the algorithms and ii) $\theta$ i.e. fidelity threshold passed as input to the algorithms. The latter would provide a conservative estimate of our bounds since, $f^* \geq \theta$ for a certified region. We can see that adaptI converges to probability 1 the fastest in terms of $Q$, probably because of its ability to hone in on the low fidelity regions leading to higher values of the cdf and hence tighter bounds. In terms of time, unif is the fastest since we just have to estimate a single cdf for each region we certify. For adaptI, approximately $\log(Q)$ cdfs have to be estimated per region, while for unifI it is $\approx q(1 + \log\log(Q))$, which leads to the higher time. Nonetheless, in practical terms, all bounds seem to be reasonably efficient to compute (within $\sim 6$ minutes here).

| Explanation method | Criterion | Strategies | $f^*$ proxy | $Q$ | | | |
| | | | | 10 | 100 | 1000 | 10000 |
|---|---|---|---|---|---|---|---|
| LIME | time (s) | unif | $\hat{f}^*$ | 0.1350 | 0.1750 | 0.2250 | 0.7350 |
| | | | $\theta$ | 0.1289 | 0.1612 | 0.2023 | 0.7011 |
| | | unifI | $\hat{f}^*$ | 0.1543 | 1.4000 | 13.9179 | 373.9050 |
| | | | $\theta$ | 0.1399 | 1.2234 | 12.7832 | 365.1237 |
| | | adaptI | $\hat{f}^*$ | 0.1620 | 0.4200 | 0.8100 | 3.8220 |
| | | | $\theta$ | 0.1494 | 0.4012 | 0.7914 | 3.5822 |
| | bounds | unif | $\hat{f}^*$ | 0.0000 | 0.4833 | 1.0000 | 1.000 |
| | | | $\theta$ | 0.0000 | 0.4665 | 0.9892 | 1.000 |
| | | unifI | $\hat{f}^*$ | 0.0002 | 0.7746 | 1.0000 | 1.000 |
| | | | $\theta$ | 0.0001 | 0.7582 | 0.9862 | 1.000 |
| | | adaptI | $\hat{f}^*$ | 0.0002 | 0.9668 | 1.0000 | 1.000 |
| | | | $\theta$ | 0.0002 | 0.9589 | 1.0000 | 1.000 |
| SHAP | time (s) | unif | $\hat{f}^*$ | 0.1470 | 0.1860 | 0.2340 | 0.8110 |
| | | | $\theta$ | 0.1298 | 0.1821 | 0.2218 | 0.7981 |
| | | unifI | $\hat{f}^*$ | 0.1656 | 1.5200 | 14.1200 | 375.3340 |
| | | | $\theta$ | 0.1581 | 1.4827 | 12.2371 | 369.4216 |
| | | adaptI | $\hat{f}^*$ | 0.1710 | 0.5100 | 0.8900 | 3.9870 |
| | | | $\theta$ | 0.1601 | 0.4897 | 0.8691 | 3.7456 |
| | bounds | unif | $\hat{f}^*$ | 0.0000 | 0.4550 | 0.9840 | 1.0000 |
| | | | $\theta$ | 0.0000 | 0.4417 | 0.9612 | 1.0000 |
| | | unifI | $\hat{f}^*$ | 0.0003 | 0.7844 | 1.0000 | 1.0000 |
| | | | $\theta$ | 0.0001 | 0.7519 | 0.9831 | 1.0000 |
| | | adaptI | $\hat{f}^*$ | 0.0003 | 0.9384 | 1.0000 | 1.0000 |
| | | | $\theta$ | 0.0003 | 0.9272 | 1.0000 | 1.0000 |

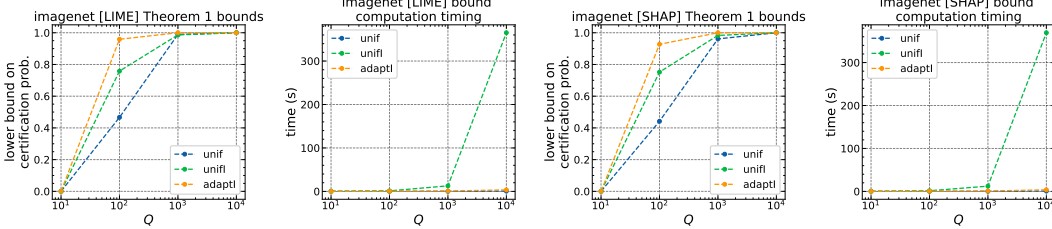

Figure 9: Visualizing trends for the results in Table 7, where $f^*$ proxy is $\theta$ (conservative estimate). Results are qualitatively similar when $f^*$ proxy is $\hat{f}^*$. Left two figures are bounds and timing for LIME respectively. Right two figures are bounds and timing for SHAP respectively.

