# OpenReview forum: "Trust Regions for Explanations via Black-Box Probabilistic Certification"
_ICLR.cc/2024/Conference — Submitted to ICLR 2024_

### Official Review · Reviewer_3Uus · 2023-10-29

**Soundness:** 3 good
**Presentation:** 3 good
**Contribution:** 2 fair
**Rating:** 5
**Confidence:** 4

**Summary:**

The authors present a formal approach to "explanation certification" in machine learning. Given a model with query access, a sample point, and a set quality metric, they aim to identify the largest hypercube centered at the sample point. Within this hypercube, any applied explanation should meet the set quality criterion. To achieve this, they introduce "Ecertify," a method that samples different points around the sample point within a hypercube's width and evaluates them based on the quality metric. The size of the hypercube are then iteratively adjusted until the maximum size is identified. The authors introduce several sampling techniques - 'unif', 'unifI', and 'adaptI', and provide probabilistic guarantees on their effectiveness.

**Strengths:**

The paper introduces a novel problem of explanation certification and highlight the benefits, i.e., insight into model behavior in a specific region, stability of explanation, reusing explanations, etc.

They provide rigorous experimental  analysis to support their findings.

**Weaknesses:**

While the problem statement is compelling, the methodology employed seems somewhat heuristic. Given a metric \(f(\cdot)\), the goal is to identify a hypercube wherein all points satisfy the requirement \(f(\cdot) > \theta\). The approach involves sampling points within the hypercube, querying the function, and subsequently adjusting the hypercube's width size. The probabilistic guarantees suggest that as the number of queries approaches infinity, the error diminishes with high probability. This is reminiscent of the Maximum Likelihood Estimation (MLE) of a uniform distribution \([0,b]\) using samples \( \max_i x_i \). Here, \(\hat{b} < b\), and as \(i\) tends towards infinity, one approaches \(b\) at an exponential rate. What differentiates one sampling method from another if all have similar convergence rate?

The paper seems to overlook the role of the quality function \(f\) in its analysis. The provided probabilistic guarantees necessitate knowledge of \(f's\) cumulative distribution function (CDF), which is unknown. Nonetheless, the paper's discussion of special cases, such as characterizing CDFs for linear models and enhancing certification for Lipschitz models, is intriguing. However, relegating this information to the appendix is a missed opportunity. I recommend that the authors emphasize these aspects more prominently in the main body of the paper.

Identifying a hypercube in a d-dimensional space may lack practical relevance, especially since features in real-world datasets aren't always uniformly scaled.

Additionally, there's existing literature on robust counterfactual explanations amidst noisy implementations [1] and other related works that focus on identifying a region where a model's prediction remains consistent. These studies also employ a similar strategy of sampling around a point and querying the model. While their formulations might differ from yours, it's essential to acknowledge and cite these works

The presentation of the methodology in the paper, especially regarding 'unifI' and 'adaptI', could benefit from further clarity. Incorporating a visual representation or figure summarizing the methods might be beneficial. The inclusion of the video in the supplementary material is commendable.

[1] https://arxiv.org/pdf/2203.06768.pdf

**Questions:**

Address weaknesses section.
Generally, I find your problem interesting, however I find your methods to be heuristic (sample and query; probabilistic guarantees are directly from a law of large numbers style argument). I would expect for a method a leverage the quality function $f$ for a better solution. I might be missing something and would like the authors to clarify the non-triviality of their method. I am open to revising my score .

---

> ### Author Response · Authors · 2023-11-15
> **Response to reviewer 3Uus**
>
> > What differentiates one sampling method from another if all have similar convergence rate? I would expect for a method to leverage the quality function $f$ for a better solution.
>
> As mentioned in Section 5, although the rate of convergence is exponential in $Q$ for all three strategies, we see that for unifI (eq. 4) and adaptI (eq. 5) the exponent heavily depends on the cdfs of the quality function $f$ around the prototypes that are sampled. Hence, if we have even one good prototype that gets sampled, both unifI and adaptI will leverage this and perform better than unif. adaptI goes a step further and samples (exponentially) more around the promising prototypes. Hence, our methods do leverage $f$ and the benefit of this is seen in the experiments (see Figure 2), where adaptI is faster than all the other methods since it is able to hone in on the low fidelity examples. Moreover, looking at Tables 5 and 7 in the appendix where we report the computed bounds based on Theorem 1, we observe that the bounds for adaptI converge faster than for unifI, which in turn converge faster than for unif (see especially $Q=100$). These observations are also mentioned in the experimental section.
>
> > The paper seems to overlook the role of the quality function (f) in its analysis... I recommend that the authors emphasize these aspects (characterizing CDFs for linear models and enhancing certification for Lipschitz models) more prominently in the main body of the paper.
>
> As mentioned in our previous response, our bounds do not overlook the quality function $f$, and in fact heavily depend on the specific sampling strategy paying close attention to where it samples. We are glad that you liked our analysis of the special cases. We have pointed to this section in the introduction and related work. In terms of adding more material from the appendix to the main body, we would be grateful if you could suggest what could be removed as given the tight page limit of 9 pages and having now added also a figure in the introduction (see our last response below), we are finding it challenging to make space for it. In the main body as it stands, we chose to focus more on cdf estimation and cdf-free bounds from extreme value theory in Section 6.1, as these approaches apply more widely than the special cases.
>
> > Identifying a hypercube in a d-dimensional space may lack practical relevance, especially since features in real-world datasets aren't always uniformly scaled.
>
> As mentioned in the discussion, our approach could also be used to find hyper-rectangles with some additional book-keeping. The relative lengths could correspond to the non-uniform scalings. Moreover, given the scalings one could rescale when applying our algorithm so that spreads are uniform and scale back when outputting the regions. Additionally, if the data lies on a lower dimensional manifold, one could apply our methods in the latent space (e.g. latent space of an autoencoder) and then decode to obtain free-form regions in the input space. This is also mentioned in Appendix G.
>
> > there's existing literature on robust counterfactual explanations amidst noisy implementations [1] and other related works... While their formulations might differ from yours, it's essential to acknowledge and cite these works
>
> We have now cited such works in the related works section. Thank you for pointing us to them.
>
> > 'unifI' and 'adaptI', could benefit from further clarity. Incorporating a visual representation or figure summarizing the methods might be beneficial.
>
> We have now added Figure 1 to the paper which tries to convey the main ideas behind the strategies. More detailed description is provided in Section 4. We did this to bring the benefit of the supplementary videos that you appreciated into the main paper.

---

> > ### Comment · Reviewer_3Uus · 2023-11-23
> >
> > Thank you for your rebuttal. I have reviewed your response along with other reviews and rebuttals. I will keep my current score.

---

> > > ### Author Response · Authors · 2023-11-23
> > > **More Clarifications?**
> > >
> > > Thank you for your response. We would be happy to provide further clarifications for any outstanding issues you might still have. Although the discussion phase is ending soon we would be happy to engage. Thank you.

---

### Official Review · Reviewer_pTb7 · 2023-10-31

**Soundness:** 2 fair
**Presentation:** 1 poor
**Contribution:** 2 fair
**Rating:** 3
**Confidence:** 4

**Summary:**

This paper introduces a black box explanation certification method to provide insight into model behavior and ensure explanation stability. Empirical evaluations on synthetic and real data demonstrate the effectiveness of the proposed work. This work seems to have solid theoretical analyses and deals with an interesting topic. However, the work is not well-written. In particular, some intuition and concepts are not clearly provided, making the work difficult to follow.

**Strengths:**

1. The work deals with an interesting topic, i.e., finding a hypercube of samples to make the explanations stable for the samples within the hypercube.
2. The authors provide rich theoretical analyses to support their claim.

**Weaknesses:**

1. The design of the methodology is quite trivial. The authors only provide the pseudo-code of the method without clear explanations and descriptions at a high level, which makes the reviewer unable to get the intuition behind the idea.
2. Although the research topic is interesting, the motivation is not strong and it is unclear how important stability is for the XAI. Is there a practical scenario to illustrate the significance of the research topic?
3. It is unclear the pros and cons of the three sampling strategies. The authors did not provide a deeper discussion of the three sampling strategies.
4. The experiments are confusing. What is the purpose of ZO and why is it used as a baseline? Why does the convergence of w to a similar value indicate the accuracy of the explanation?

**Questions:**

1. What is the intuition of method design? At the least, the author should provide one figure to illustrate why the method is designed in this way and how the method works.
2. Could the authors provide a real-world application to demonstrate the significance of the research?
3. What are the pros and cons of the three sampling strategies? What kind of scenario are they suitable for and what insight do they give practitioners?
4. The authors claimed the method is model agnostic and quite general. So can the method generalize to graph data?
5. To what extent is the found w stable? i.e., the bound of P (f (x);w) >=theta.

---

> ### Author Response · Authors · 2023-11-15
> **Response to reviewer pTb7**
>
> > What is the intuition of method design? At the least, the author should provide one figure to illustrate why the method is designed in this way and how the method works.
>
> We have now added Figure 1 to the paper which tries to convey the main ideas behind the strategies. More detailed description is provided in Section 4. Also note that we had uploaded videos showing the working of the three strategies in our original submission.
>
> > Could the authors provide a real-world application to demonstrate the significance of the research?
>
> One real-world application is *explanation reuse*. As mentioned in (Dhurandhar et al., 2019) (and also in our experience), given today's cloud-driven world where a model could exist on one cloud platform and explainability is offered as a service by a different corporation, each query to the black box model has an associated cost and hence finding explanations for many examples in a dataset can be prohibitive, not just in terms of time and network traffic but also monetarily. Finding regions where an existing explanation is valid can produce significant savings in all these three aspects, since one now does not have to find explanations for every example which typically takes $1000$s of queries (e.g. $5000$ is the default for LIME).
>
> Another example is in finance where banks want to find cohorts where a certain action (related to credit cards, loans, etc.) might have similar effect on all members of the cohort. Our approach will provide such cohorts where a certain explanation is valid for the cohort and hence recourse may be possible. More applications can be found in  (Liao et al., 2022), where they surmise that stability of explanations, which will lead to such trust regions, is important for stakeholders performing tasks such as model improvement, domain learning, adapting control and capability assessment.
>
> > What are the pros and cons of the three sampling strategies? What kind of scenario are they suitable for and what insight do they give practitioners?
>
> As mentioned in the Experiments section, the pro of unif is that it is the simplest and also sometimes the fastest strategy. unifI and adaptI are more complicated but are preferable when you have $1000$s or $10000$s of features respectively. Our advice would be to use unif for up to $100$s of features, unifI for $1000$s of features, and adaptI for $10000$s of features or greater.
>
> > The authors claimed the method is model agnostic and quite general. So can the method generalize to graph data?
>
> Yes, the method can work for graph data if the graphs are embedded in some compact space. For example by vectorizing adjacency matrices or using embeddings from Graph Neural Networks (GNNs).
>
> > To what extent is the found $w$ stable? i.e., the bound of $P (f (x);w) >=\theta$.
>
> As seen in the experiments (see Tables 2-4 in the appendix), the found $w$ is quite stable from $Q=1000$. Even $Q=100$ is sufficient for low dimensions. The bounds are also quite tight from $Q=1000$ (see Tables 5, 7 in the appendix).
>
> > What is the purpose of ZO and why is it used as a baseline?
>
> As mentioned in the related work, ZO is a standard approach used for black-box adversarial attacks. We adapted this method to our setup since ours is a novel problem in XAI with no prevalent baselines, where we thought an adapted version of ZO would be the closest and fairest baseline.
>
> > Why does the convergence of $w$ to a similar value indicate the accuracy of the explanation?
>
> First note that the convergence of $w$ relates to the accuracy of the estimated *trust region width* for a given threshold $\theta$ and has nothing to do with the quality or accuracy of an *explanation*. For the synthetic experiments, we know the true $w$ and hence we can validate the different methods with certainty. For the real datasets, the true $w$ are unknown. Thus the different methods converging to similar values gives some confidence that the methods are returning an accurate enough $w$, as the mechanics behind the methods are quite different (especially between ZO and the others) and hence their converging to a similar value just by chance is unlikely.

---

> > ### Comment · Reviewer_pTb7 · 2023-11-23
> > **Response**
> >
> > Thank you for your rebuttal. I have reviewed your response along with other reviews and rebuttals. I will keep my current score.

---

### Official Review · Reviewer_KmSN · 2023-11-01

**Soundness:** 4 excellent
**Presentation:** 3 good
**Contribution:** 3 good
**Rating:** 6
**Confidence:** 4

**Summary:**

This paper proposed a novel certificate of explanability in a hyper-ball centered at a given sample that is model-agnostic. The authors provide rigorous formulation, solutions with theoretical guarantees and analysis, and experiments on many datasets.

**Strengths:**

1.	The novel formulation, adopting from certificate for adversarial robustness, is very interesting and could potentially be applied in practice
2.	The analysis and explanation of the proposed methodologies, described in Algorithm 1 is very clear and comprehensive.
3.	The theoretical analysis of the performance guarantees of Algorithm 1 and 2 is novel and useful for different strategies to explore the regions
4.	Numerical results are abundant, including large-scale images datasets and synthetic datasets.

**Weaknesses:**

1.	Despite that the authors mentioned in Section 8 Discussion that the l_\infinty ball can potentially be generalized to hyper-rectangles or l_p balls, all these hyper-balls may not carry semantic meanings of the samples. Therefore even if we can certificate that the pseudo-samples within the hyper-balls have high fidelity, those pseudo-samples may not carry meaningful information, since human interpretation on images is very different from Euclidean distances. It is encouraged that the authors address this concern with a discussion.
2.	The connection between the Algorithm 1 and Algorithm 2 proposed in Section 4 seems to be detached from the original optimization formulated in equation 1. It is encouraged that the authors strengthen the connection.
3.	The authors mention several advantages of the novel explainability certificate, including stability of the explanations and explanation reuse, but did not provide numerical results or analysis on these benefits over existing explainable AI tools. It is encouraged that the authors
4.	This certificate is closely related to adversarial examples, and in related literature there are also studies on the maximal hyper-balls that will not suffer from a decision change. Their algorithm is also similar to the searching algorithm proposed in Algorithm 1. The novelty here is the quality metric and the explanation function. It is encouraged that the authors include more discussion on different explanation functions, or give concrete examples in Section 2.
5.	In XAI, the explainability of features is sometimes more important than individual samples. Is it possible to extend the techniques to features?
6.	One of the most important aspects in XAI tools is visualization. It is encouraged that the authors add visualization of the regions, provide some intuitive explanations, with simple (semi-)synthetic data.

**Questions:**

Please refer to the Weaknesses. I will consider raising the scores if the authors could adequately address my questions in the rebuttal.

---

> ### Author Response · Authors · 2023-11-15
> **Response to reviewer KmSN**
>
> > Despite that the authors mentioned in Section 8 Discussion that the $l_\infty$ ball can potentially be generalized to hyper-rectangles or $l_p$ balls, all these hyper-balls may not carry semantic meanings of the samples. Therefore even if we can certificate that the pseudo-samples within the hyper-balls have high fidelity, those pseudo-samples may not carry meaningful information...
>
> Yes you are right that all examples in a region may not be realistic input examples in the domain lying on the data manifold. In such cases one could after the region is found consider only realistic examples in the region for their downstream task. In a sense we would provide a superset of examples for which the explanation is valid.
>
> Another option as discussed in Appendix G is one could simply apply our methods to the latent space
> (e.g. as learned by an auto-encoder) rather than the input space. Thus, although the regions will
> be hypercubes in the latent space they will be more free-form in the input space which might be interesting. As such, our approaches should still apply.
>
> > The connection between the Algorithm 1 and Algorithm 2 proposed in Section 4 seems to be detached from the original optimization formulated in equation 1.
>
> Algorithm 1 decides on a width $w$ and Algorithm 2 tries to certify that width (i.e. check if $f_{x_0}(x) \ge \theta \quad \forall x \in B_\infty(x_0, w)$). If certified, Algorithm 1 chooses a bigger width which again Algorithm 2 tries to certify. Thus, the two algorithms are tightly linked to the optimization formulated in eq. 1, where we are trying to find the maximum possible (certified) width.
>
> > It is encouraged that the authors include more discussion on different explanation functions, or give concrete examples in Section 2.
>
> Example explanation functions would be linear like in LIME, logistic regression, small decision trees, rule lists, interpretable neural architectures such as CoFrNet. We have now mentioned some of these in Section 2.
>
> > Is it possible to extend the techniques to features?
>
> A thing to note here is that when we find trust regions we are finding instances in the input space where an explanation, which are typically feature importances, is valid
> for all of them. In other words, the feature importances hold for all examples in that region. Hence, we are highlighting important features in that region.
>
> However, if your goal is to find features that are similar one could potentially transpose the data matrix and find regions of similar features, where the explanations would now point to important instances.
>
> > It is encouraged that the authors add visualization of the regions, provide some intuitive explanations, with simple (semi-)synthetic data.
>
> We have now added Figure 1 to the paper which tries to convey the main ideas behind the strategies. More detailed description is provided in Section 4. Also note that we had uploaded videos showing the working of the three strategies in our original submission.
>
> > The authors mention several advantages of the novel explainability certificate, including stability of the explanations and explanation reuse, but did not provide numerical results or analysis on these benefits over existing explainable AI tools.
>
> Our approach compliments the current work in XAI where XAI toolkits (viz. AIX360, Captum, InterpretML, etc.) contain predominantly explanation methods and/or (quality) metrics. One can take these methods and metrics and apply our approach on top to obtain trust regions, which could be used for any of the applications outlined in the introduction. As such in the experiments we do provide a comparison between LIME and SHAP using our methodology that is of a different flavor than previous works. This type of analysis can be used to compare and contrast XAI methods
> on individual examples, on regions, as well as on entire datasets, and across different models. Also note that our methods perform quite well around $Q=1000$ where we obtain a region (i.e. a set of examples) in which an explanation is valid. This is (potentially) much fewer queries than say LIME-like methods which by default query a model $5000$ times per example it wants to find an explanation for.

---

> > ### Comment · Reviewer_KmSN · 2023-11-22
> >
> > I appreciate the authors for their response. I will keep my current score.

---

> > > ### Author Response · Authors · 2023-11-22
> > > **Thank you**
> > >
> > > We are glad that our responses were satisfactory to you.

---

### Official Review · Reviewer_46DG · 2023-11-01

**Soundness:** 3 good
**Presentation:** 2 fair
**Contribution:** 2 fair
**Rating:** 3
**Confidence:** 2

**Summary:**

In the explainability problem, we are given a classifier model and an input, and we seek to find an explanation for the classification. This paper considers the problem of finding a certificate for explanations with respect to some input quality metric. The certificate(also called a trust region) is the largest hypercube centered at the example such that at least 1-\delta of the points in the hypercube meet the quality criteria. The trust region has multiple applications, including explanation reuse and analysis of model behavior.

The paper first introduces the problem formally and then presents three techniques for certification, the simplest of which involves uniform sampling to determine the density of points that violate the quality metric. The other techniques refine the random sampling strategy for faster convergence.  The algorithms are implemented, and the experiments highlight that the certification technique is faster than comparable methods while providing roughly similar trust regions. An interesting insight from the experiments is that LIME trust regions can be larger than SHAP if the desired fidelity is on the lower side.

**Strengths:**

Originality and Significance: The idea of certification of explanations and its potential application to explanation reuse seems to be novel and quite interesting. The experimental evaluation is relevant, and the insights are valuable.

**Weaknesses:**

I felt only the first of the three techniques is well motivated, while the other two don't really contribute to the main idea of the paper. Since the main contribution is a formalization of the novel problem, it would be nice to have a simple example to illustrate what a certification looks like.

There are also a few issues with the readability:
1) What is fidelity? Perhaps the definition is not strictly necessary for understanding the paper, but since it is used so many times, I think it is critical to define it.
2) the pseudo-code and the explanation in section 4 are not well written. I may be wrong, but the pseudo-code seems to be executing a binary search, in which case it is needlessly obscured.

**Questions:**

For w1>w2>w3, it seems to be possible that hypercubes of halfwidth w1 and w3 could be valid trust regions while w2 isn't. Since the problem statement involves finding the largest certificate, would the certification really be meaningful around the example? For instance, if the explanation passes the quality threshold over most of the domain but is bad around important points, it seems that the certificate could just be the entire domain, revealing little about the point of interest. Is my understanding correct?

---

> ### Author Response · Authors · 2023-11-14
> **Response to reviewer 46DG**
>
> > I felt only the first of the three techniques is well motivated
>
> In Section 4 last paragraph, we motivate the three strategies. As mentioned there and further referenced in Appendix D, we previously shared videos to provide more intuition about the methods. Now in the introduction we have added a figure to further clarify the working of these methods.
>
> As such, unifI ($2^{\text{nd}}$ technique) resembles a dynamic grid search where we randomly pick different locations in the current region and search around them. The $3^{\text{rd}}$ technique adaptI does a dynamic adaptive search where we again randomly pick different locations in the current region and search around them, but additionally focus our search more and more around the locations that we find more promising points (a.k.a. low fidelity examples).
>
> > it would be nice to have a simple example to illustrate what a certification looks like.
>
> As mentioned above we have now added Figure 1 to the paper which tries to convey the main ideas behind the strategies. More detailed description is provided in Section 4. Also note that we had uploaded videos showing the working of the three strategies in our original submission.
>
> Certification in Figure 1 would be our algorithm outputting $w=0.5$ with the (certification) probability based on Theorem 1 being $1$ for all three strategies.
>
> > What is fidelity?
>
> As you rightly mentioned, the exact definition of fidelity is not critical to the exposition but, to improve clarity, we now in Section 2 point to eq. 14 in the appendix, where we have defined fidelity consistent with previous works (Dhurandhar et al., 2022; 2023; Ramamurthy et al., 2020).
>
> > ... but the pseudo-code seems to be executing a binary search, in which case it is needlessly obscured.
>
> Algorithm 1 doubles or halves the width depending on if the current region was certified or not. It is similar to binary search but with subtle differences such as the doubling and also what upper bound width to set when a region is found to be invalid (i.e. the "else" statement). Algorithm 2 embodies the three strategies for certifying a fixed region and is very different than binary search.
>
> > For w1>w2>w3, it seems to be possible that hypercubes of halfwidth w1 and w3 could be valid trust regions while w2 isn't.
>
> This is not possible. We are trying to find a connected (indeed, convex) region in the input space around a point where its explanation is valid for all examples within this region (with high probability). If w2 is found to be invalid our approach will return only w3 in this case.
>
> **A respectful request to reconsider:** We see that you expressed relatively low confidence (2) in your initial assessment. We hope that we have addressed what seems to be your main comment about having an illustrative example to clarify and contrast the three algorithms. We have also answered your other questions (fidelity, binary search, w1>w2>w3). In light of our response and the nature of your comments, we hope that you will reconsider your rating of 3 (reject). Many thanks in advance.

---

### Official Review · Reviewer_ot88 · 2023-11-01

**Soundness:** 2 fair
**Presentation:** 1 poor
**Contribution:** 3 good
**Rating:** 8
**Confidence:** 4

**Summary:**

This work addresses an interesting question in the area of robust explanations: How do we find a region around a point such that an explanation remains more or less unchanged? Additionally, they enforce a constraint that one only has black-box access to the function f(x) which computes the fidelity of the local explanation of x0 at x and the evaluation of the model g(x).

The paper formalizes the problem statement. Then, they provide a strategy for addressing this problem that adaptively expands a bounding box around the point using samples in that region. They provide theoretical guarantees on the performance of their proposed strategy using probabilistic methods. They also provide experimental results on several datasets to demonstrate the efficacy of their approach.

**Strengths:**

The problem is an interesting one from a mathematical standpoint.
An algorithm is proposed to find robust regions along with theoretical guarantees.
Introducing an adaptive strategy is a good idea.
Experiments have been performed on a variety of datasets which include image and tabular datasets, as well as two popular explanation techniques namely LIME and SHAP.

It is interesting to connect it back to Lipschitz functions and also piecewise linear functions in the Appendix.

**Weaknesses:**

There is insufficient motivation for this problem. For instance, why would one care for the explanations to remain the same in a region? For instance, if the model itself is undulating in a region, why should local explanations remain stable? The width of the robust region found would accordingly be large or small. How would this be informative? The motivation could be strengthened. Perhaps, a toy example could be useful.

The term black-box access is unclear since here they are not only accessing the model output but also its local explanation function. I would think black-box access typically means just computing the model output g(x), but additional functions are being computed here. So, calling it black-box access is a bit misleading.

To certify a single point itself, one requires many queries. Then, to certify multiple points, the query would significantly increase. Could the authors comment further on this computational complexity?

Also, in the experiments, it seems that this certification is being done for just one point (?) How do the results vary when trying to certify different points? Could you discuss more on the average over the entire dataset?

The experiments also do not show a uniform trend. What does that mean in this context? The experimental section needs some clarity on what should one expect to see.

The presentation of the algorithm can be significantly improved with much more clarity. The paper is quite difficult to read. Not much intuition is provided.

Also, closely related works in the area of counterfactual explanations (and some of the references therein) also talk about robust regions for counterfactual explanations.
[1] Finding Regions of Counterfactual Explanations via Robust Optimization: https://arxiv.org/pdf/2301.11113.pdf
[2] Robust Counterfactual Explanations for Neural Networks With Probabilistic Guarantees: https://arxiv.org/pdf/2305.11997.pdf
[3] Consistent counterfactuals for deep models: https://arxiv.org/abs/2110.03109

**Questions:**

Questions are provided along with the weaknesses.

---

> ### Author Response · Authors · 2023-11-14
> **Response to reviewer ot88 1/2**
>
> > There is insufficient motivation for this problem. For instance, why would one care for the explanations to remain the same in a region?
>
> As mentioned in the third paragraph of the introduction, there are multiple motivations for this work. We elaborate on some of them here.
>
> **Explanation reuse:** As mentioned in (Dhurandhar et al., 2019), given today's cloud-driven world where a model could exist on one cloud platform and explainability is offered as a service by a different corporation, each query to the black box model has an associated cost and hence finding explanations for many examples in a dataset can be prohibitive, not just in terms of time and network traffic but also monetarily. Finding regions where an explanation is valid can produce significant savings.
>
> **Insights for different stakeholders:** In (Liao et al., 2022)'s award winning work it was surmised through (AI and domain expert) user studies that stability of explanations is particularly important in recourse for stakeholders performing tasks such as model improvement, domain learning, adapting control and capability assessment. This is because without stability actions may produce unintended outcomes. Of course, as you mention, depending on the degree of non-linearity of the black box model, the sizes of the regions in which these explanations are valid will change, but this is precisely where approaches like ours would be useful in informing the user of how broadly the provided explanation would be applicable.
>
> **Meta-metric:** Given a quality metric (viz. fidelity, stability, etc.) that the user cares about, our approach could be used to ascertain which explanation method obeys the quality metric better, which would correspond to larger trust regions (on average) returned by our method.
>
> > The term black-box access is unclear since here they are not only accessing the model output but also its local explanation function.
>
> Access to the model is only through querying, a.k.a. black-box access as it is called in the literature. The local explanation function is typically separate from the model (e.g. LIME, SHAP, etc.). In fact, it could even be just feature attributions provided through some unknown process (viz. domain knowledge from a subject matter expert). Our approach works irrespective of how the explanation is obtained, and thus accesses not only the (black-box) model but also the explanation function only through querying. This justifies in our opinion the use of the term black-box.
>
> > ... to certify multiple points, the query would significantly increase. Could the authors comment further on this computational complexity?
>
> We see from the experiments that $Q=1000$ typically gives accurate enough regions. Also note that once we obtain a region, we would not certify other points within that region since we already know that the computed explanation is valid for them. This would reduce the number of queries. More importantly, LIME-like methods (by default) query around $5000$ times to obtain an explanation for a single point and are heavily used despite this cost. In contrast, our approach could potentially find an explanation for a bunch of points (those lying in the region) with say $Q=1000$.
>
> > How do the results vary when trying to certify different points? Could you discuss more on the average over the entire dataset?
>
> In Appendix F we report results with more points. The results will of course vary depending on the point and the behavior of the black-box model around it. *The key point here is not that the region sizes will vary but that we have provided a mechanism to find these regions for a wide range of explanation methods, quality metrics (with their thresholds) and black-box models.* The averages over a dataset will again depend on the non-linearity of the black-box model, the stability of the explanation method and the threshold set for the quality metric. For instance, for FICO using Boosted Trees, we find that for $\theta=0.75$ LIME produces larger regions than SHAP (on average) and hence is possibly more favorable at this threshold, but for a larger threshold such as $\theta=0.9$ SHAP is more favorable.
>
> > The experimental section needs some clarity on what should one expect to see.
>
> As mentioned in the first line of Section 7, we are trying to show through the experiments that our methods recover accurate regions and do so more efficiently than say adapted ZO approaches. We are also trying to show that our bounds derived in Section 5 are computable (see Tables 5,6,7 in the appendix). As discussed in the last paragraph of the "Observations" section, we also show how our approach can be used to compare explanations methods.

---

> > ### Author Response · Authors · 2023-11-14
> > **Response 2/2**
> >
> > > The presentation of the algorithm can be significantly improved... not much intuition provided
> >
> > We have now added Figure 1 to the paper which tries to convey the main ideas behind the strategies. More detailed description is provided in Section 4. Also note that we had uploaded videos showing the working of the three strategies in our original submission.
> >
> > > closely related works in the area of counterfactual explanations...
> >
> > We have now cited these in the related works section. Thank you for pointing out those to us.

---

> > > ### Comment · Reviewer_ot88 · 2023-11-22
> > > **Increased Score**
> > >
> > > Based on the responses, I have increased my score to Accept.

---

> > > > ### Author Response · Authors · 2023-11-22
> > > > **Thank you**
> > > >
> > > > We are glad our responses were satisfactory to you.

---

### Author Response · Authors · 2023-11-14
**Common Response**

We thank the reviewers for their constructive comments and efforts taken in reviewing our manuscript. We are glad that you found our work to be **novel** and **interesting** with the experimental section being **valuable**, **abundant** and **rigorous** supplemented with **rich theoretical analysis**. Based on the reviews we have made the following changes to the paper:

1. We have added a figure (Figure 1) to clarify the main ideas behind the three strategies (unif, unifI and adaptI).

2. Added references on robust counterfactual explanations in related work.

3. Made other minor additions based on reviewer suggestions.

4. To make space for the above and maintain the 9 page limit we have moved to the appendix (previously) proposition 1 (now proposition 2) and most of the discussion section which even previously was predominantly in Appendix G.

Note that all additions to the paper are highlighted in **blue**. We now individually address each reviewer's concerns.

---

### Author Response · Authors · 2023-11-19
**Request to consider our rebuttal below**

Dear Reviewers,

We are thoroughly thankful for your initial reviews. Based on your reviews we have made changes to the paper and have also tried to alleviate your main concerns through our rebuttal below (posted Nov 14th). We would be ever more grateful if you could consider our response before the author-reviewer discussion phase ends on Nov. 22nd in case there are more questions we need to clarify.

Thank you,
Authors

---

### Meta-Review · Area_Chair_umji · 2023-12-10

**Metareview:**

This paper proposed a new method to build and certify the stability of explanations. The proposed methods aim to characterize the stability of individual examples through trust regions – i.e., regions of feature space over which an example remains stable. The paper formalizes this problem and develops model-agnostic algorithms to construct trust regions. Their proposed approach is validated is theoretical guarantees as well as experiments on synthetic and real-world datasets.

**Strengths**

- Flexible Framework
- Sound Techniques to Construct Trust Regions

**Weaknesses**

- Lack of Use Cases & Demonstrations - The paper solves a technical problem but lacks a compelling use case. This point was brought up by multiple reviewers who asked how trust regions could be used in practice. The authors have included a list of plausible use cases - but should include demonstrations that their method adds value to these real-world applications. This is especially important since producing trust regions for explanations – can inadvertently induce the use of explanations that are stable but ineffective or incorrect. This could help practitioners decide what kinds of explainability techniques  deciding what kinds of explainability techniques to choose for  arise because of explanations

**Justification For Why Not Higher Score:**

See review.

**Justification For Why Not Lower Score:**

N/A

---

### Decision · Program_Chairs · 2024-01-16

Reject